# Fatty acid desaturation and lipoxygenase pathways support trained immunity

Anaísa V. Ferreira [1,2,7] ✉, Juan Carlos Alarcon-Barrera[3,7],
Jorge Domínguez-Andrés [1], Özlem Bulut[1], Gizem Kilic[1], Priya A. Debisarun [1],
Rutger J. Röring [1], Hatice N. Özhan [1], Eva Terschlüsen[4], Athanasios Ziogas[1],
Sarantos Kostidis [3], Yassene Mohammed [3], Vasiliki Matzaraki[1],
George Renieris [5], Evangelos J. Giamarellos-Bourboulis [5],
Mihai G. Netea [1,6] & Martin Giera [3] ✉

Infections and vaccines can induce enhanced long-term responses in innate immune cells, establishing an innate immunological memory termed *trained immunity*. Here, we show that monocytes with a trained immunity phenotype, due to exposure to the Bacillus Calmette-Guérin (BCG) vaccine, are characterized by an increased biosynthesis of different lipid mediators (LM) derived from long-chain polyunsaturated fatty acids (PUFA). Pharmacological and genetic approaches show that long-chain PUFA synthesis and lipoxygenase-derived LM are essential for the BCG-induced trained immunity responses of human monocytes. Furthermore, products of 12-lipoxygenase activity increase in monocytes of healthy individuals after BCG vaccination. Grasping the underscoring lipid metabolic pathways contributes to our understanding of trained immunity and may help to identify therapeutic tools and targets for the modulation of innate immune responses.

Adaptive immunity is characterized by the capacity to retain memory to a specific antigen, however, a growing body of evidence also attributes memory features to the innate immune system– a property termed *trained immunity*. Epidemiological studies have shown that the tuberculosis vaccine BCG increases innate immune cell function and confers non-specific protection to heterologous infections, thus decreasing all-cause mortality[1]. Innate immune cells, such as monocytes, after brief exposure to BCG or the fungal cell wall component β-glucan, exhibit a long-term augmented production of pro-inflammatory cytokines in response to a secondary non-related stimulation[2,3].

Trained immunity is supported by epigenetic modifications that alter gene transcription and metabolic rewiring of diverse metabolic pathways. The metabolic changes of trained immune cells provide energy, signaling molecules, and cellular components that impact innate immune effector functions. Various studies have determined the contribution of glycolysis, the TCA cycle, and oxidative respiration to the establishment of the trained immunity phenotype[4–6]. However, the contribution of other metabolic pathways in the context of trained immunity remains to be fully explored, as is the case of pathways related to lipid metabolism. Several clues from literature point towards the possible relevance of lipid metabolism pathways in trained immunity. For example, cholesterol biosynthesis is increased in hematopoietic stem and progenitor cells of β-glucan trained mice[7], and the metabolite mevalonate, involved in the cholesterol synthesis pathway, amplifies trained immunity in human monocytes[8].

[1]Department of Internal Medicine and Radboud Center for Infectious Diseases (RCI), Radboud University Nijmegen Medical Center, 6500HB Nijmegen, The Netherlands. [2]Instituto de Ciências Biomédicas Abel Salazar (ICBAS), Universidade do Porto, 4050-313 Porto, Portugal. [3]Center for Proteomics and Metabolomics, Leiden University Medical Center, 2333ZA Leiden, the Netherlands. [4]Department of Medical Microbiology, Radboud University Medical Centre, 6500HB Nijmegen, The Netherlands. [5]4th Department of Internal Medicine, National and Kapodistrian University of Athens, Medical School, Athens, Greece. [6]Department for Immunology and Metabolism, Life and Medical Sciences Institute (LIMES), University of Bonn, 53115 Bonn, Germany. [7]These authors contributed equally: Anaísa V. Ferreira, Juan Carlos Alarcon-Barrera. ✉e-mail: anaisa.validoferreira@radboudumc.nl; m.a.giera@lumc.nl

Lipids and lipid mediators (LM) could act as signaling molecules to establish trained immunity. LM are bioactive signaling molecules produced through the oxidation of polyunsaturated fatty acids (PUFA) by the action of cyclooxygenases (COX), lipoxygenases (LOX), and epoxygenase (CP450) enzymes. The selective engagement of these pathways allows for a tailored LM response to different inflammatory states[9]. For example, eicosanoids, prostaglandins (PGs), and leukotrienes (LTs) are classically related to a proinflammatory activity, enhancing innate and adaptive immune functions. However, it has become accepted that the resolution of acute inflammation is also an active process, regulated, in part, by specialized lipid mediators such as resolvins, maresin, protectins, and lipoxins[10]. Moreover, recent studies have also started highlighting the tissue regenerative and anti-inflammatory properties of prostaglandin $E_2$ ($PGE_2$)[11].

In this study, we hypothesize that LM influences the induction of trained immunity. We combine lipidomic, genomic, and pharmacological approaches to investigate whether the induction of trained immunity in monocytes by BCG relies on the modulation of LM biosynthesis and composition. We show that human monocytes exposed in vitro to BCG are enriched for different LM. We identify genetic variations in fatty acid desaturases (*FADS*) and *LOX* genes that are associated with the trained immunity phenotype, as mirrored by the increase of cytokine production capacity upon BCG vaccination. Furthermore, we observe that 12-LOX-derived LM in monocytes are increased in individuals vaccinated with BCG.

## Results

### Monocytes trained in vitro with BCG display increased concentrations of lipoxygenase-derived metabolites

Trained immunity inducers, such as β-glucan and BCG, trigger long-term metabolic reprogramming of monocytes and macrophages. However, the role of LM in the induction of trained immunity has not yet been comprehensively investigated. We used an established in vitro model of trained immunity[12], in which human peripheral blood monocytes were exposed to β-glucan or BCG for 24 h and left to rest for the following 5 days in the absence of any stimulus. In contrast, to induce immune tolerance, monocytes were exposed to LPS on day 0 of culture.

We investigated changes in LM content at two different time points: after 24 h of treatment and at day 6 of culture, in the absence of restimulation (Fig. 1, S2, S3). The concentration of free PUFA arachidonic acid (AA), eicosapentaenoic acid (EPA) and docosahexaenoic acid (DHA) was not altered by the different stimulations at the evaluated timepoints. However, despite donor variation, we observed clear trends in the concentrations of different LM at 24 h of exposure (Fig. 1). BCG triggered the increased production of eicosanoids and docosanoids, primarily associated with 12- and 15-LOX activity, namely 15-HETE and 12-HETE (Fig. 1a), 15-HEPE, 12-HEPE (Fig. 1b), and 14-HDHA (Fig. 1c). 5-HETE, a product of 5-LOX activity was also enriched in BCG-exposed monocytes (Fig. 1a). $PGE_2$, a product of COX activity, was likewise significantly increased in BCG-exposed monocytes (Fig. 1a). Interestingly, recent publications on 15-hydroxyprostaglandin dehydrogenase (15-PGDH) inhibitors highlight the role of $PGE_2$ on cell and tissue regeneration[13,14]. Overall, BCG stimulated an increase in the biosynthesis of LM, particularly 12- and 15-LOX products. In contrast, LPS promoted the depletion of most species, particularly 12-LOX derived products as observed by the significantly decreased concentrations of 12-HETE, 12-HEPE and 14-HDHA, in comparison with BCG-stimulated monocytes. The LM enrichment induced by BCG was triggered early on during monocyte stimulation and were not persistent after the differentiation of trained cells at day 6 of culture. In contrast, at day 6, LPS-stimulated cells tended to show decreased levels of different LM, such as 17-HDHA, 15-HEPE, and 5-HEPE (Fig. S2, S3). In accordance, a previous study compared human monocytes and differentiated macrophages

and observed a decrease in LM synthesis capacity and release in differentiated cells, possibly pinpointing an exhaustion of PUFA precursor pools (Fig. S2, S3)[15].

### Pharmacological inhibition of polyunsaturated fatty acid desaturase 2 (FADS2) decreases BCG-induced trained immunity in vitro

Monocytes exposed to BCG for 24 h exhibit a general increase in concentrations of PUFA-derived LM. We then hypothesized that these pathways and the activity of fatty acid desaturases, COX, and LOX enzymes play a role in the establishment of BCG-induced trained immunity. We applied an in vitro pharmacological approach, in which monocytes were exposed to different enzymatic inhibitors concomitantly with BCG stimulation for 24 h.

Firstly, we evaluated the importance of fatty acid desaturation. We assessed the expression of the main desaturases stearoyl-CoA desaturase (SCD) and fatty acid desaturases 1 and 2 (FADS1/2). The expression of *SCD* did not appear to be modulated at 4 h or 24 h after BCG exposure, while *FADS1* tended to be decreased and *FADS2* tended to be increased at 4 h (Fig. 2a). FADS2 is responsible for desaturation of short-chain PUFA, such as the essential linoleic acid and alpha-linolenic acid precursors, into long-chain PUFA, like AA, DHA and EPA. Pharmacological inhibition of FADS2 by SC-26196 decreased LPS-induced TNFα and IL-6 production in BCG-trained macrophages (Fig. 2b, c). Curiously, inhibition of FADS2 increased TNFα production in unstimulated macrophages, contrary to the downregulation seen in BCG-trained cells.

Epidemiological evidence points to the role of PUFA and macrophages in the development of inflammatory conditions such as ulcerative colitis and Crohn's disease[16,17]. Also, a large number of studies suggest the involvement of mycobacteria in the pathophysiology of Crohn's disease[18], and one may speculate that inappropriate induction of trained immunity by mycobacteria in the gut mediates this relationship. Thus, we set out to explore this possibility by means of FADS2 inhibition in a mouse model of colitis. DSS treatment induced inflammation, as is reflected in the increase of DAI score over time, while concomitant inhibition of FADS2 led to a significant decrease in DAI score at day 6 (Fig. S4A−C), while no significant changes in weight loss at the same time point were observed (Fig. S4D). The decrease of BCG-induced responsiveness by FADS2 inhibition led us to interrogate the Liver X receptors (LXR). LXR are nuclear receptors that provide transcriptional control of lipid metabolism genes. Particularly, macrophage LXR activation was shown to enhance PUFA synthesis and *FADS1/2* gene expression[19]. Similarly to FADS2 inhibition, pharmacological inhibition of LXR also decreased BCG-enhanced TNFα and IL-6 production after LPS stimulation (Fig. 2d). Jointly, these results underpin the importance of long-chain PUFA for the development of trained immunity.

### LOX-derived LM increases monocyte responsiveness in vitro

We next explored the role of COX- and LOX- activity in the establishment of trained immunity. The inhibition of the COX pathway did not ablate BCG-induced enhanced cytokine production (Fig. 3a). Although expression of *COX2* was increased in monocytes 4 h following BCG treatment (Fig. 3b), pharmacological inhibition of COX-1, COX-2, or CP450 did not decrease TNFα or IL-6 secreted upon LPS restimulation. Instead, aspirin increased the production of TNFα in both control and BCG-trained monocytes (Figs. 2b, 3a). Accordingly, pharmacological inhibition of prostanoid receptors EP2-4 or DP2 did not modulate BCG-induced cytokine production (Fig. S5A). Interestingly, exogenous supplementation of monocytes with the COX product $PGE_2$, which was increased upon BCG 24 h stimulation, decreased TNFα and IL-6 production after LPS restimulation (Fig. 3c, d).

In contrast, inhibition of 5-LOX or 12-LOX enzymes decreased BCG-enhanced cytokine production (Figs. 2b, 4a). 5-LOX inhibition

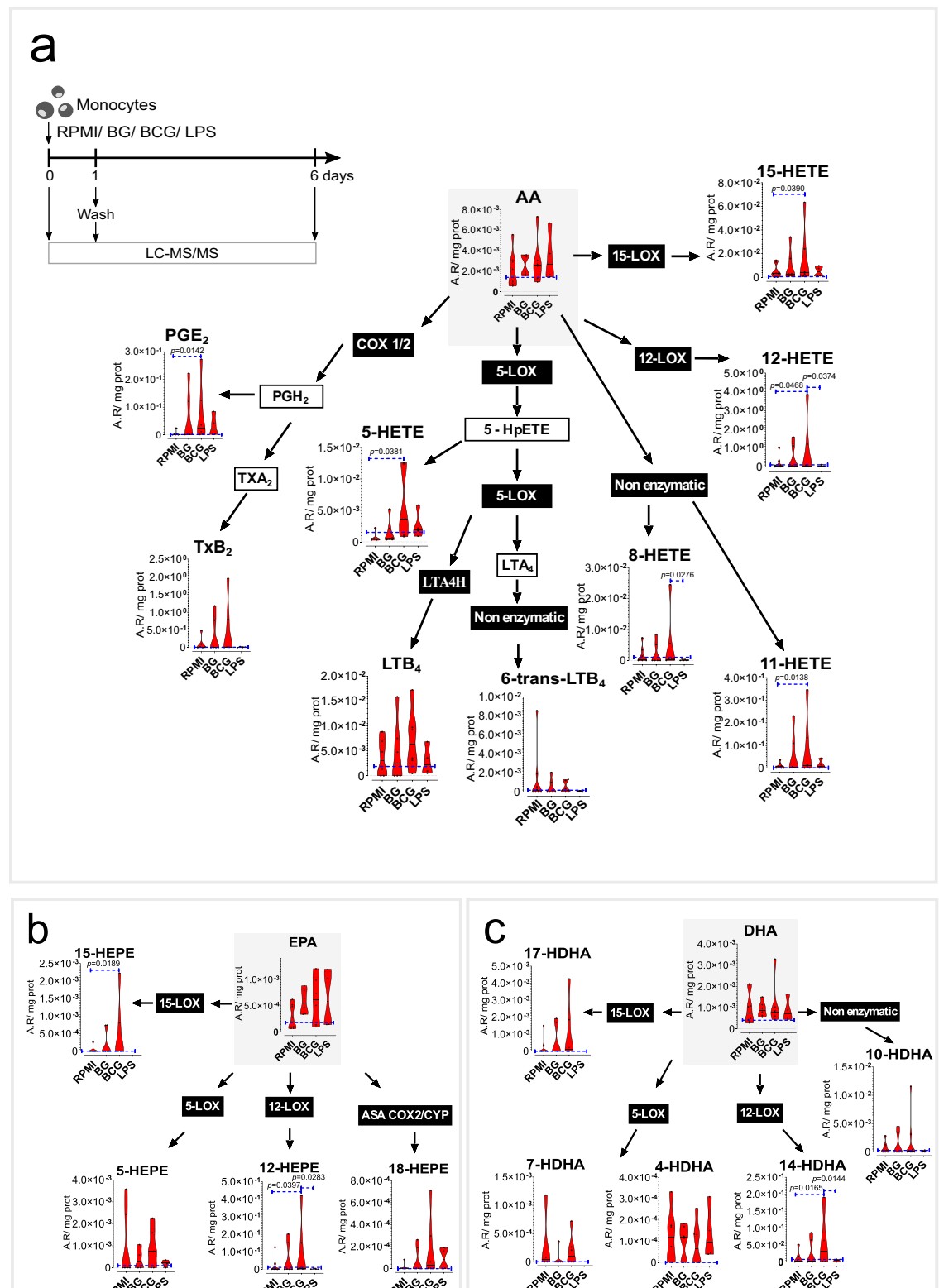

**Fig. 1 | Increased PUFA biosynthesis in BCG-trained monocytes is associated with eicosanoid and docosanoid biosynthesis after 24 hours.** Levels of PUFA and oxylipids in stimulated monocytes at 24 h with BG, BCG, and LPS (n = 4-8 biologic replicates, pooled from 4 independent experiments), **a** Arachidonic acid pathway (AA), **b** Eicosapentaenoic acid (EPA), and **c** Docosahexaenoic acid (DHA) pathway. Violin plots showing median and the interquartile region, Friedman's test

followed by Dunn's multiple comparisons test. Area ratios (A.R), corrected for internal standards and protein content are shown. (Ctrl control, BG β-glucan, BCG Bacillus Calmette-Guérin, LPS lipopolysaccharide, LOX lipoxygenase, COX cyclooxygenase, PG prostaglandin, Tx thromboxane, LT leukotriene, HETE hydroxyeicosatetraenoic acid, HEPE hydroxyeicosapentaenoic acid, HDHA hydroxy docosahexaenoic acid).

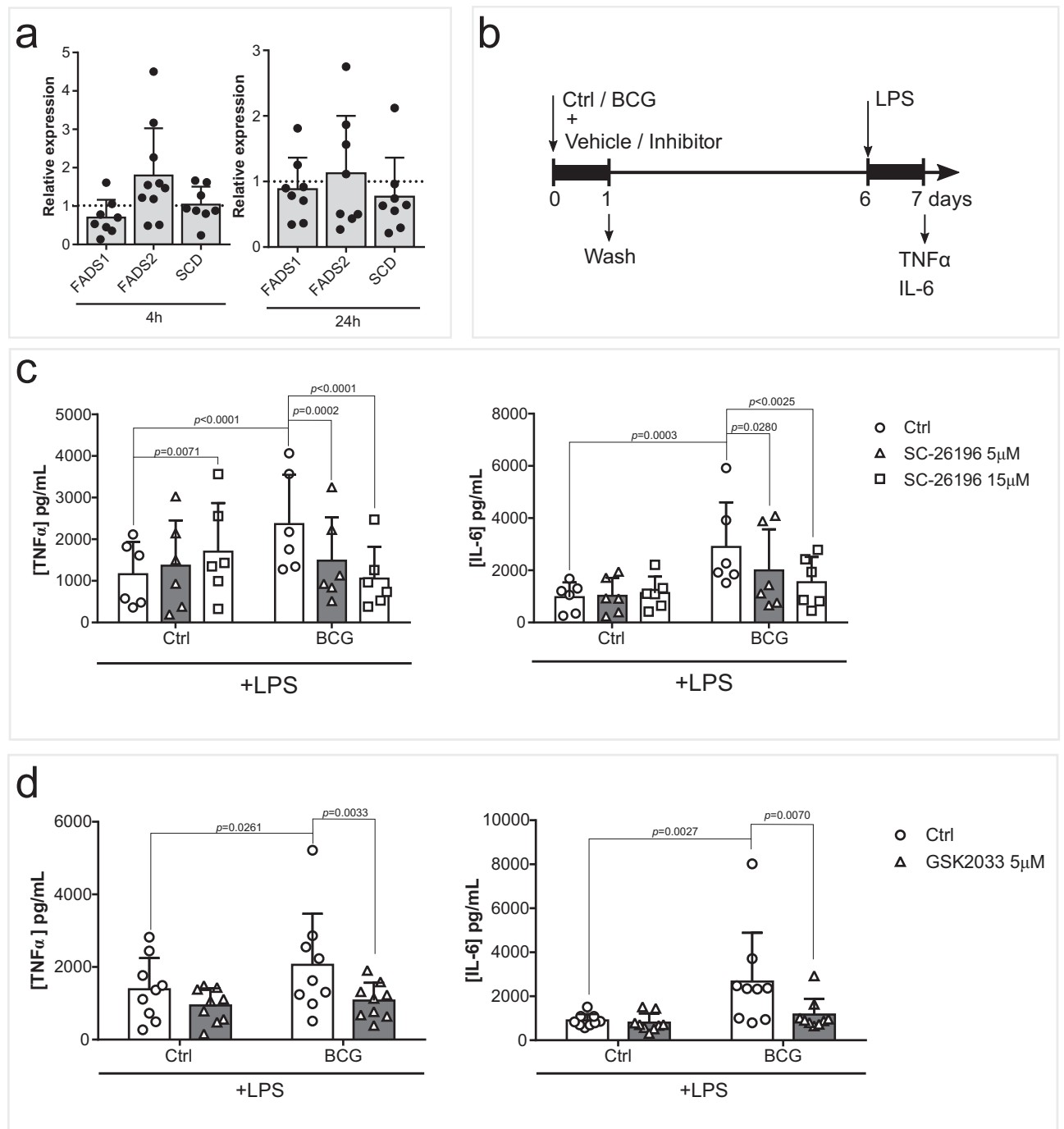

**Fig. 2 | FADS2 activity contributes to the induction of trained immunity.**
**a** Monocyte *FADS1, FADS2, SCD* gene expression 4 h and 24 h after BCG exposure (n = 8, except for FADS2 at 4 h which is n = 10 biologic replicates). **b** Schematic representation of the in vitro model used for the pharmacological inhibition of the induction of BCG-trained immunity. **c** Production of TNFα and IL-6 by BCG-trained macrophages incubated with (5 μM, 15 μM) SC-26196 for the first 24 h of culture and restimulated at day 6 with LPS for 24 h. (n = 6 biologic replicates pooled from 2 independent experiments). **d** Production of TNFα and IL-6 by BCG-trained macrophages incubated with (5 μM) GSK2033 for the first 24 h and restimulated with LPS at day 6 of culture for further 24 h. (n = 6 biologic replicates pooled from 2 independent experiments). Mean + SD. **c, d** two-way ANOVA, Sidak's multiple comparisons test. (Ctrl: control; BCG: Bacillus Calmette-Guérin; LPS: lipopolysaccharide; FADS: fatty acid desaturase SCD: stearoyl-CoA desaturase).

decreased TNFα and IL-6 production, while 12-LOX inhibition decreased IL-6 secretion of BCG-trained macrophages. On the other hand, 15-LOX inhibition potentiated IL-6 production of BCG-trained macrophages and increased TNFα secretion of control cells. Following these observations, we assessed whether exogenous supplementation of different LOX-derived LM is sufficient to increase responsiveness to secondary stimulation (Figs. 3c, 4b–d). Monocytes were exposed to

5-HETE, 7-HDHA, and leukotriene B₄ (LTB₄), products of 5-LOX activity (Fig. 4b); 15-HETE and 17-HDHA, products of 15-LOX (Fig. 4c); 12-HETE, 12-HEPE, and 14-HDHA, product of 12-LOX (Fig. 4d) for 24 h and left to differentiate for 5 days. 7-HDHA, 15-HETE, 17-HDHA, 12-HETE and 12-HEPE supplementation triggered increased TNFα and IL-6 production upon LPS restimulation of differentiated macrophages. Although LTB₄, 5-HETE and 14-HDHA were enriched in BCG exposed monocytes,

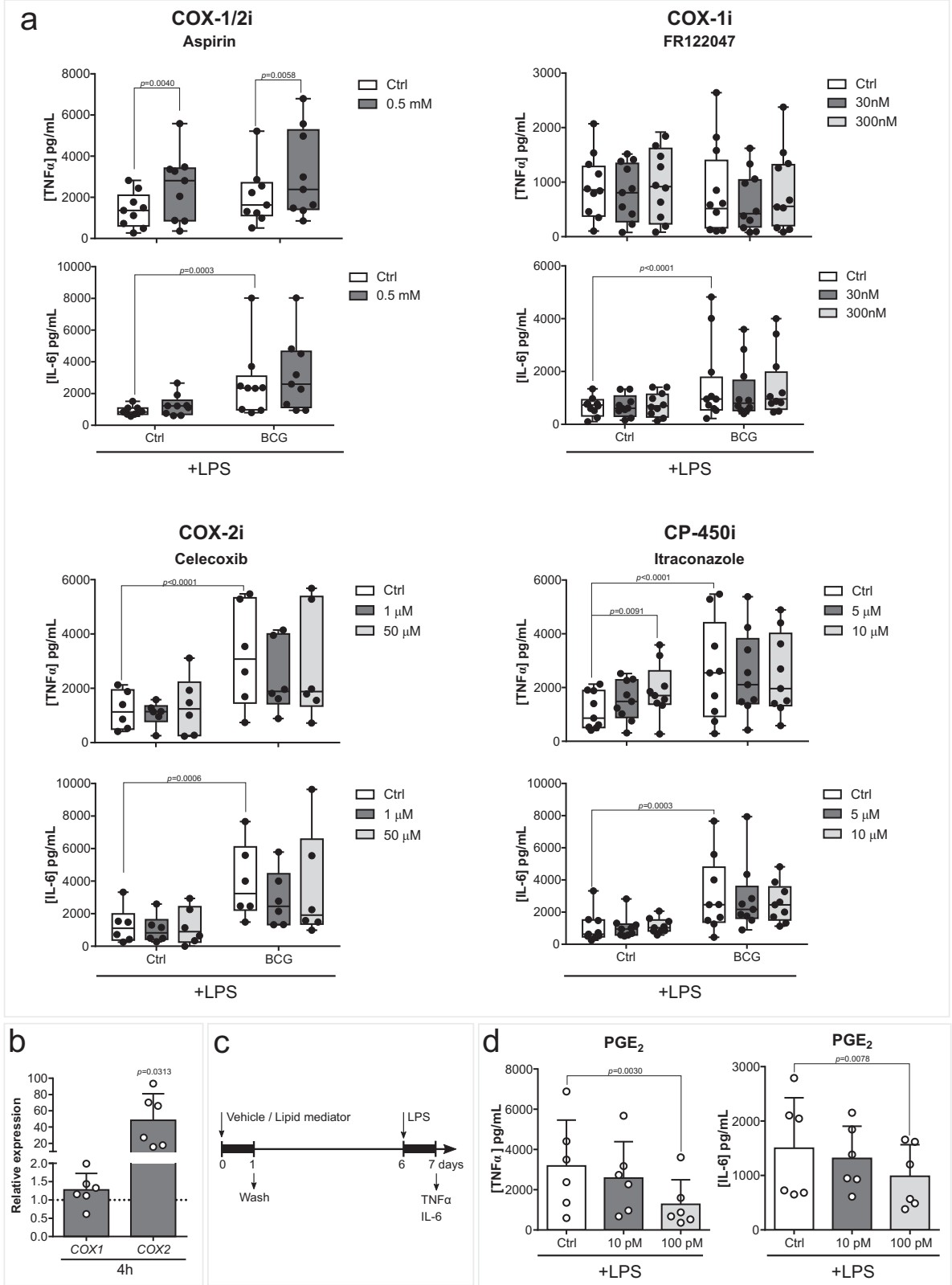

exogenous supplementation of these lipids did not modulate cytokine production upon restimulation (Fig. 4b–d). It is important to highlight that 15-LOX inhibition had an inverse proportion with cell responsiveness while supplementation with 15-LOX products proved to be positively correlated with cytokine production. This discrepancy might be due to off-target effects of the employed 15-LOX inhibitor, or other not metabolic product-related effects of 15-LOX as recently

identified for example for 12-LOX[20]. Moroever, we did not use combinations of 15-LOX products in our study.

Taken together, the COX pathway does not appear to play a significant role in the increased responsiveness of BCG-trained macrophages, while LOX-derived LM generally increase cell responsiveness to restimulation. In line with these results, the inhibition of 5-LOX and 12-LOX activity ablates BCG-enhanced cytokine production, thus

**Fig. 3 | Inhibition of the COX pathway does not decrease BCG-induced cytokine production. a** TNFα and IL-6 secreted by macrophages exposed to BCG and (0.5 mM) Aspirin, (30, 300 nM) FR122047, (1, 50 μM) Celecoxib or (5, 10 μM) Itraconazole for the first 24 h of culture and 5 days later stimulated with LPS for 24 h (*n* = 9 (Aspirin and Itraconzole), n = 12 (FR122047), *n* = 6 (Celecoxib) biologic replicates, pooled from 3 (Aspirin and Itraconazole), 4 (FR122047) or 2 (Celecoxib) independent experiments). **b** Relative *COX1* and *COX2* expressions in monocytes incubated with BCG for 4 h (*n* = 6 biologic replicates pooled from 2 independent experiments). **c** Schematic representation of in vitro LM supplementation experiments. **d** TNFα and IL-6 secreted by macrophages exposed to (10, 100 pM) PGE$_2$ for the first 24 h of culture and 5 days later stimulated with LPS for 24 h (*n* = 6 biologic replicates, pooled from 2 independent experiments). **a** Mean, 10–90 percentile and whiskers extend to the most extreme points, **b, d** Mean + SD. **a** two-way ANOVA, Sidak's multiple comparisons test, **b** two-tailed Wilcoxon matched-pairs signed rank test, **d** Friedman's test followed by Dunn's multiple comparisons test. (Ctrl control, BCG Bacillus Calmette-Guérin, LPS lipopolysaccharide, COX cyclooxygenase, CP cytochrome P, PG prostaglandin).

pointing to the relevant role of the LOX pathways for the establishment of trained immunity.

## SNPs in desaturases and LOX pathway genes influence the induction of trained immunity in healthy human volunteers

To further explore the role of desaturases and LOX pathways in trained immunity, we performed a genetic study in a cohort of 325 healthy volunteers. We assessed whether common SNPs (MAF > 0.05) in a window of 250 kb around the genes *FADS1*, *FADS2*, *SCD*, *LOX5*, *LOX15* and *LOX12* were associated with changes in pro-inflammatory cytokine production. We performed this analysis in two different models of trained immunity, an in vitro β-glucan or BCG trained immunity model, and also in an in vivo BCG vaccination model of trained immunity (Fig. 5a, b). Taking advantage of these two models, we identified various SNPs within 250 kb of the genes of interest that were suggestively associated ($p < 9.99 \times 10^{-3}$) with changes in TNFα, IL-6, and IL-1β production. The most strongly associated SNPs in each model and training stimuli are highlighted in the scatter plots. Specifically, the variation in *FADS2* (rs9326418, *p* = 0.00016) was associated with the increase of TNFα production after in vitro β-glucan-induced trained immunity, and *LOX12* (rs12232535, *p* = 0.00021) was associated with the potentiation of IL-6 in the in vitro BCG trained immunity model (Fig. 5a). Similarly, variations in *FADS2* (rs11600019, *p* = 0.00078) and *LOX12* (rs955461, *p* = 0.00032) were associated with the increased fold change of TNFα and IL-6, respectively, after BCG vaccination (Fig. 5b).

In addition to the role of the enzymes that synthetize LM, we also focused on the receptors of LM. Using the in vivo BCG vaccination trained immunity model, we found suggestive associations ($p < 9.99 \times 10^{-3}$) between changes in TNFα, IL-6, and IL-1β production and SNPs in the vicinity (± 250 kb) of genes coding for receptors of different prostanoids (*PTGDR2*, *PTGER1-3*, *PTGIR*, *TBXA2R* and *LTB4R*) (Fig. S5B) and oxylipins (*CMKLR1*, *FPR2*, *GPR18*, *GPR31*, *GPR32*, *GPR37*, *LGR6* and *OXER1*) (Fig. S6).

## 12-LOX derived LM are increased in monocytes of healthy individuals after BCG vaccination

Next, we sought to translate our in vitro findings to an in vivo setting. We investigated the concentrations of AA-, EPA- and DHA- derived LM in monocytes from healthy individuals 30 days after BCG vaccination (Fig. 5c, S7). Remarkably, 12-HEPE, and 14-HDHA, which are 12-LOX products of EPA and DHA, respectively, were significanlty increased after BCG vaccination (Fig. 5c), while the AA derived 12-HETE approached a significant increase. This enrichment in 12-LOX products upon BCG vaccination is thus in agreement with our in vitro observations and highlights a potentially important role of 12-LOX products for in vivo BCG-induced trained immunity in monocytes.

## Discussion

Innate immune memory is dependent on the rewiring of different metabolic pathways, such as glycolysis, glutaminolysis, and oxidative phosphorylation[21]; however, the role of lipid metabolism in trained immunity has yet to be comprehensively explored. In this study, we investigated the modulation of monocyte lipid content upon induction of trained immunity and, more specifically, the role of LM in the induction of trained immunity (Fig. S8).

We show that the long-chain PUFA biosynthesis pathway is relevant for BCG-induced trained immunity. This is in accordance with in vivo studies in which peritoneal macrophages of mice infected with BCG had higher PUFA content and arachidonic acid levels than non-infected animals[22]. Curiously, bone marrow progenitor cells of β-glucan trained mice had lower levels of PUFA and AA than the control group[7], pointing to potential differential regulation of lipid metabolism by progenitor cells when compared to peripheral monocytes. We observed that long-chain PUFA synthesis is necessary for establishing BCG-dependent trained immunity, as inhibition of fatty acid desaturation by FADS2 decreased BCG's effect on pro-inflammatory cytokine production.

FADS2 inhibition also showed beneficial effects possibly related to trained immunity in a colitis mouse model. Inflammatory bowel disease (IBD) has a common genetic burden with mycobacterial diseases[23], and non-tuberculous mycobacteria have been suggested to play a role in its etiology[24–26]. As mycobacteria are strong inducers of trained immunity[27–29] one may hypothesize that inappropriate induction of trained immunity is involved in the pathophysiology of IBD. In addition, in patients with ulcerative colitis, the intestinal mucosa presents increased concentrations of AA and AA-derived LM, such as PGE$_2$ and 12-HETE, and decreased DHA and EPA concentrations. The increased AA-derived metabolites were positively correlated with pro-inflammatory cytokine production[30]. Indeed, we show that FADS2 inhibition decreased both in vitro BCG-increased responsiveness and colitis severity in a murine model. Also, in both in vitro and in vivo trained immunity cohorts, we have identified associations between genetic variations in *FADS2* gene and pro-inflammatory cytokine production upon heterologous stimulation.

On the other hand, BCG may also induce the modulation of long-chain PUFA-derived LM via the regulation of LXRs. Macrophage LXRα expression was shown to be induced by a NOD2 ligand[31], which is a relevant intracellular receptor in the host response elicited by mycobacteria[32] and for BCG-induced trained immunity[3]. Indeed, LXR activity supports BCG-induced trained immunity since LXR antagonism blocks BCG-induced inflammatory responses to TLR2 restimulation[33]. In this line, we also observed a decrease in the response to TLR4 restimulation upon LXR inhibition of BCG-trained macrophages. LXRs are regulators of both lipid metabolism and inflammation. LXR decreases inflammatory responses since its activation can promote membrane lipid remodeling and increase membrane rigidity, thus blunting the TLR signaling cascade[34]. In mice exposed to LPS, LXR agonist administration led to a decrease in inflammatory gene expression[34], and LXR-deficient macrophages show a defect in efferocytosis and a more pronounced pro-inflammatory phenotype[35]. The impact of LXR signaling on macrophage phenotype appears to be context-dependent but it is an attractive possible regulator of LM synthesis, necessary for the induction of BCG-trained immunity.

Downstream metabolism of long-chain PUFA involves the transformation into bioactive signaling lipids through oxidation by the action of COX, LOX, CP450, and non-enzymatic pathways. Thromboxane and PGs, derived from AA through COX activity, are released by myeloid cells and modulate their migration, differentiation, and effector function[36]. *C. albicans* and BCG increase COX-2 expression and PGE$_2$ production[37–40]. Indeed, acute stimulation of human monocytes

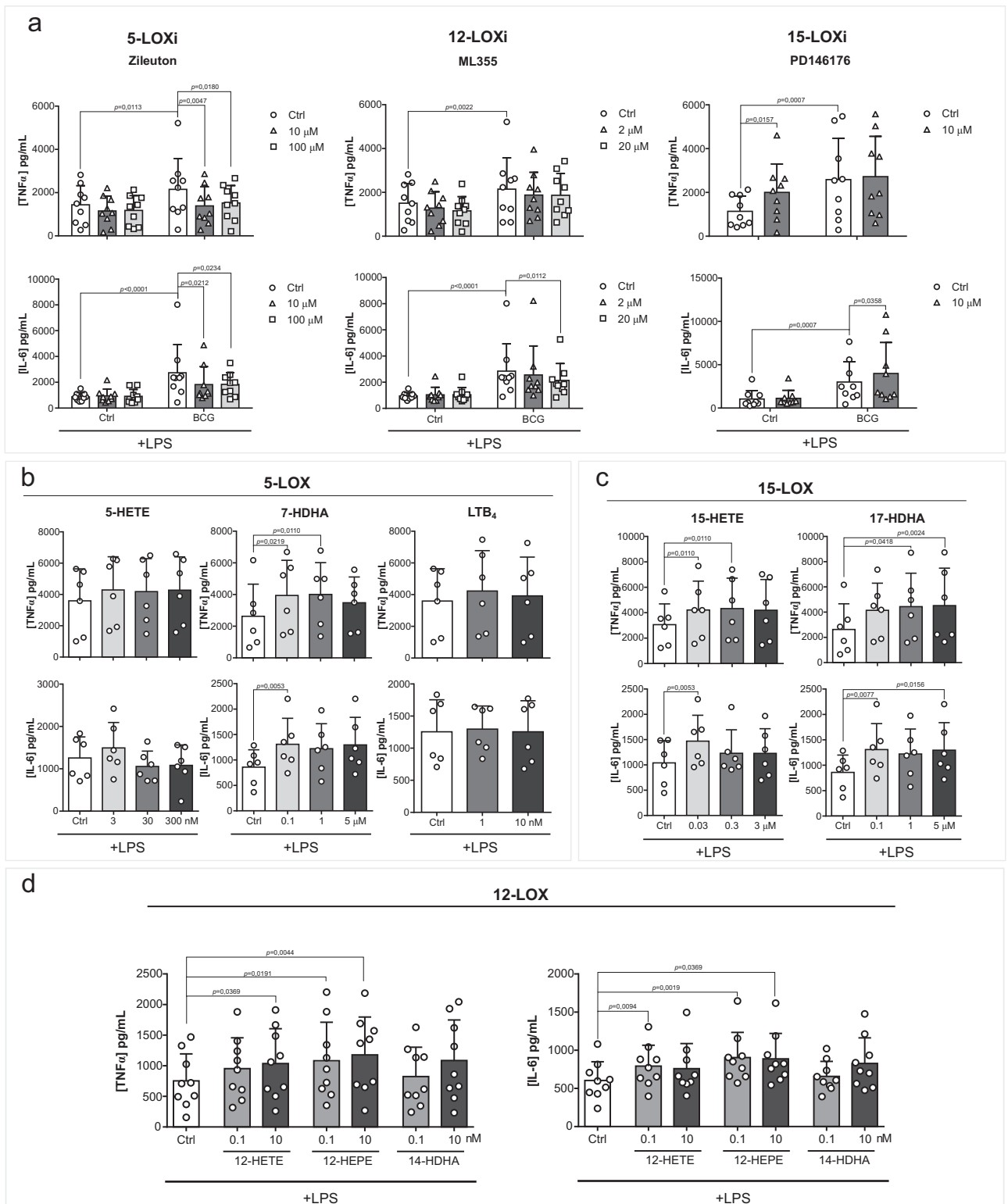

**Fig. 4 | LOX pathways play a role in BCG-induced trained immunity.**
**a** Production of TNFα and IL-6 by BCG-trained macrophages incubated with (10, 100 μM) Zileuton, (2, 20 μM) ML355 or 10 μM PD146176 for the first 24 h and restimulated with LPS for another 24 h at day 6 of culture. (*n* = 9 biologic replicates, pooled from 3 independent experiments). **b–d** Production of TNFα and IL-6 by macrophages incubated with **b** (3, 30, 300 nM) 5-HETE, (0.1, 1, 5 μM) 7-HDHA, (1, 10 nM) LTB₄, (**c**) (0.03, 0.3, 3 μM) 15-HETE, (0.1, 1, 5 μM) 17-HDHA, **d** (0.1, 10 nM)

12-HETE (0.1, 10 nM) 12-HEPE or (0.1, 10 nM) 14-HDHA for 24 h and restimulated at day 6 of culture with LPS for 24 h. (*n* = 6 (**b, c**) or *n* = 9 (**d**) biologic replicates, pooled from 2 (**b, c**) or 3 (**d**) independent experiments). Mean + SD, **a** two-way ANOVA, Sidak's multiple comparisons test, **b–d** Friedman test followed by Dunn's multiple comparisons correction. (Ctrl: control; BCG: Bacillus Calmette-Guérin; LPS: lipopolysaccharide; LOX: lipoxygenase).

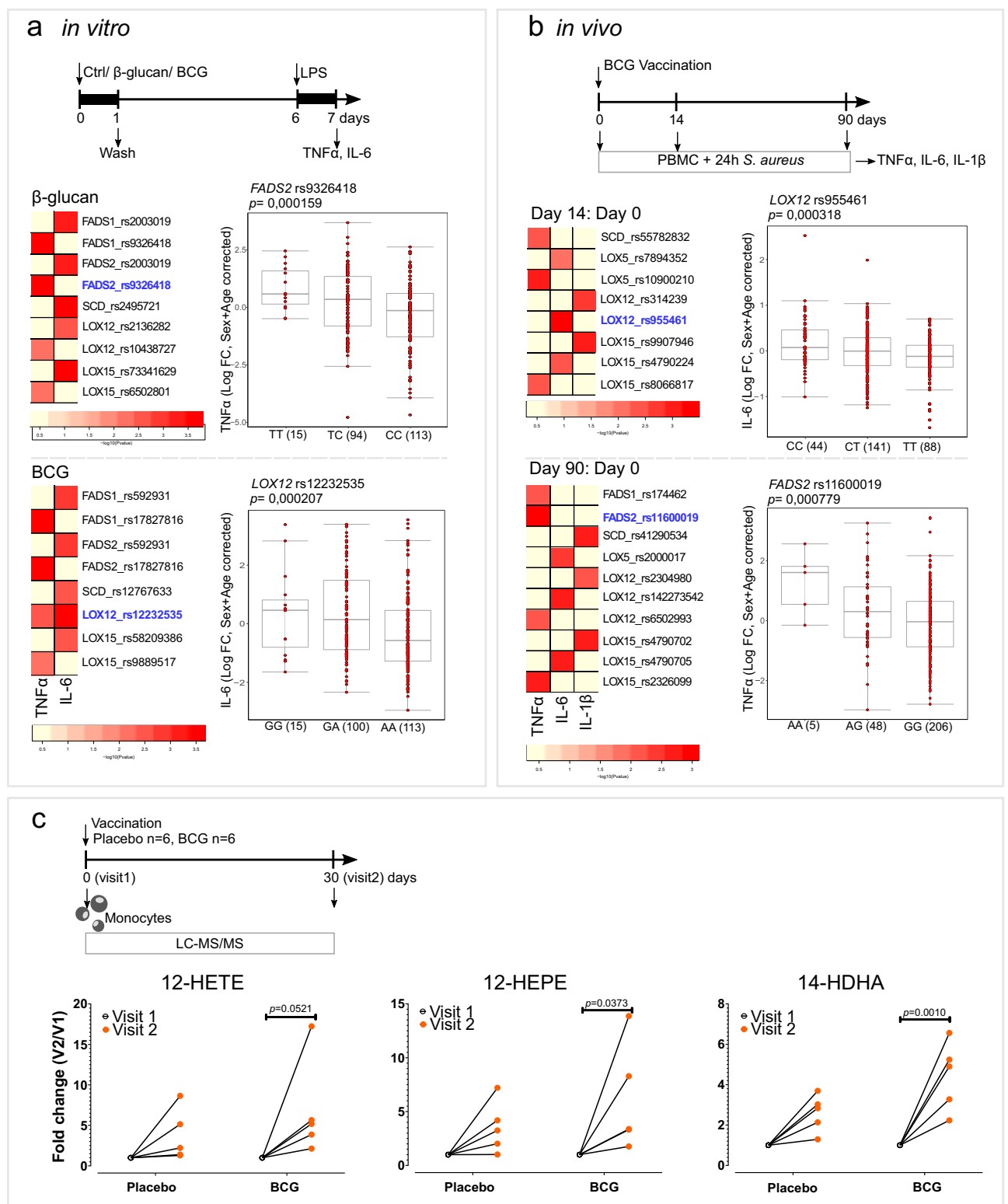

**Fig. 5 | Lipid desaturation and the LOX pathway are involved in the in vivo induction of trained immunity. a**, **b** Heatmaps of the p values of the association between SNPs around *FASD1, FADS2, SCD, LOX5, LOX12* and *LOX15* genes (+/− 250 kb) and the fold change of the production of (**a**) TNFα and IL-6 after in vitro exposure to β-glucan or BCG and LPS restimulation, (n = 238 healthy volunteers for TNFα and n = 251 healthy volunteers for IL-6) and **b** TNFα, IL-6 and IL-1β upon BCG vaccination and *S. aureus* ex vivo restimulation (n = 278 healthy volunteers). Boxplots show selected SNPs with the lowest p-values stratified according to genotype, shown in the heatmap in blue. Boxplots show median,

upper, and lower quartiles, and whiskers extend to the most extreme point less than 1.5 times the interquartile range from the box. **c** Fold change of intracellular monocyte levels of 12-LOX products of AA (12-HETE), EPA (12-HEPE), and DHA (14-HDHA) one month after BCG vaccination (visit 2) relative to levels prior to vaccination (visit 1) (n = 6 healthy volunteers per group, one outlier per group was removed (ROUT test (Q = 5%)), two-way ANOVA, Sidak's multiple comparisons test. (Ctrl control, BCG Bacillus Calmette-Guérin, LPS lipopolysaccharide, PBMC peripheral blood mononuclear cells, FC fold change, A adenine, T thymine, C cytosine, G guanine; V visit.)

with BCG increased $PGE_2$ intracellular concentrations, but after the differentiation period, $PGE_2$ concentrations were unchanged between trained and naive macrophages. However, despite this acute increase in $PGE_2$ concentrations, inhibition of COX activity or inhibition of prostanoid receptors did not modulate inflammatory responses of restimulated macrophages. In contrast, $PGE_2$ exogenous supplementation decreased pro-inflammatory cytokine production induced by LPS five days after the stimulation with the LM. This result is in line with the well-known anti-inflammatory effects of $PGE_2$[41,42] and with the concept that "the beginning programs the end" in inflammation resolution[43]. Also, trained immunity programs rely on the potentiation of glycolysis and oxidative respiration[6], and $PGE_2$ has been shown to suppress both pathways[44]. Taken together, the COX pathway plays an important immunological role but not in the induction of BCG-trained immunity.

In addition to the COX pathway, the LOX enzymes produce AA, DHA, and EPA-derived LM. We show that monocytes exposed to BCG were generally enriched in LOX products. The observed that the in vitro increase in LOX-derived mediators was corroborated by the increase in 12-LOX products in circulating monocytes of BCG-vaccinated individuals. In a BCG vaccination human cohort, we also observed suggestive associations between SNPs near oxylipins receptor genes and pro-inflammatory cytokine production upon ex vivo stimulation. Of note, the in vitro assays and the genetic study were performed with the BCG-Bulgaria strain while individuals that participated in the in vivo lipid analysis were vaccinated with BCG-Denmark strain. Importantly, both strains point to the upregulation of 12-LOX products in BCG-induced trained immunity. Curiously, the 12-LOX product 12-HETE has been suggested as a plasma biomarker for the diagnosis of TB[45], and concentrations of 5-LOX products were also increased in the plasma of multidrug-resistant TB patients[46]. In this study, eicosanoids were shown to positively correlate with the plasma concentrations of succinate and fumarate, metabolites also enriched during induction of trained immunity[5,46].

Different LOX-derived metabolites appear to be necessary and sufficient for the responsiveness of macrophages. In vitro inhibition of 5- and 12-LOX enzymes downregulated BCG-induced cytokine production, while specific LOX-derived LM were sufficient to increase responsiveness to TLR-4 stimulation, characteristic of trained immunity. However, we cannot exclude that the supplemented LM were further metabolized and thus had an indirect effect on cytokine production. LOX-derived LM is attributed anti-inflammatory and pro-resolving roles[47], but this activity is context-dependent. In a mouse model of fungal infection with *Aspergillus fumigatus*, 5-LOX-deficient mice showed impaired neutrophil recruitment to the lung and decreased survival[48]. In contrast, 5-LOX-deficient mice were more resistant to *M. tuberculosis* infection[49]. It is important to bear in mind that macrophages are highly dynamic cells, and different inflammatory and infectious conditions may trigger distinct lipid responses. For example, TLR-2 or TLR-4 stimulation promoted COX-2 and 5-LOX pathways in the pro-inflammatory M1 macrophages, while the anti-inflammatory M2 macrophages showed increased release of 12-LOX and 15-LOX products[15]. Also, LXR activation in M2 macrophages promoted the expression of 15-LOX and led to an increase of 12/15-LOX products[50]. In contrast, macrophages supplemented with the AA 12/15-LOX product 12-HETE showed increased IL-6 and $TNF\alpha$ expression[51]. Accordingly, IL-6 and $TNF\alpha$ expression was enhanced in peritoneal macrophages isolated from mice overexpressing the 12/15-LOX gene[51].

It is important to note that in humans there are three described 12/15 lipoxygenases, *ALOX12*, *ALOX15* and *ALOX15B*[52]. In macrophage populations, the expression of ALOX12 is low or even null[53]. Nevertheless, others have also shown a significant production of 12-HETE in human monocytes[21]. Moreover, the decrease of BGC-enhanced cytokine production induced by 12-LOX inhibition points to the role of this enzyme. Also, it is possible that the increase in 12-lipoxygenated products we observed in vitro and in vivo BCG-exposed monocytes is reflective of increased 15-LOX activity. 15B-LOX is constitutively expressed and produces almost exclusively oxygenated products at carbon 15; on the other hand, in addition to 15-lipoxygenated products, 15-LOX is also able to catalyze the synthesis of oxygenated products at carbon 12[54]. *ALOX15* expression is mainly enhanced in M2 macrophages[53] and in macrophages that contribute to the resolution of inflammation[55]. Yet, pharmacological inhibition of 15-LOX did not decrease BCG-enhanced cytokine production, which questions the possible role of this enzyme in the production of 12-lipoxygenated products in BCG-exposed monocytes. In summary, further studies are necessary to pinpoint the exact enzymes involved in LM production in BCG-trained immunity.

In conclusion, LM, in particular LOX products, are essential for the induction of trained immunity. LOX-derived products increased macrophage responsiveness, similar to other key metabolites in trained immunity, such as fumarate and mevalonate, while inhibition of the long-chain PUFA synthesis pathway dampened cytokine production. Furthermore, circulating monocytes of BCG-vaccinated individuals, which were previously shown to produce increased levels of pro-inflammatory cytokines upon heterologous stimulation[29], were enriched in 12-LOX products. In summary, the monocyte lipidome appears to directly influence the trained immunity phenotype, and its composition likely reflects immune memory. Further studies are warranted to decipher and correlate complex lipid species, such as esterified LOX products, with short- and mid-term immune memory as well as the exact enzymes and possible macrophage sub-population responsible for our findings. Nevertheless, this study highlights the relevance of the LOX pathway in trained immunity, and future studies should explore the LOX pathway as a therapeutic tool for the potentiation of the innate immune system or a target to decrease maladaptive inflammatory programs.

## Methods

### Isolation of human peripheral blood mononuclear cells (PBMC) and monocytes

Buffy coats from healthy donors were obtained after written informed consent (Sanquin Blood Bank, Nijmegen, the Netherlands). Isolation was performed by differential density centrifugation using Ficoll-Paque (GE Healthcare). Subsequently, the isolation of monocytes was performed with a hyper-osmotic Percoll (Sigma) density gradient centrifugation and washed once with pyrogen-free cold phosphate-buffered saline (PBS). Cells were resuspended and later cultured in RPMI 1640 Dutch modified medium (Invitrogen) supplemented with 5 μg/mL gentamicin (Centraform), 2 mM Glutamax (Gibco), and 1 mM pyruvate (Gibco). To ensure maximal purity, Percoll-isolated monocytes were left to adhere to polystyrene flat bottom plates (Corning) for 1 h at 37 °C 5%, $CO_2$ and then washed once with warm PBS. For lipid analysis, monocytes were negatively selected from PBMCs (Pan Monocyte Isolation Kit, Miltenyi Biotec) and cultured as mentioned above in supplemented RPMI 1640 Medium with no phenol red (Invitrogen).

### In vitro trained immunity model

Monocytes were cultured as previously described[12]. Briefly, monocytes were incubated with 1 μg/mL β-glucan, 5 μg/mL BCG vaccine (BCG-Bulgaria, InterVax, Canada) or 1 ng/mL *Escherichia coli* lipopolysaccharide[56] (LPS; serotype 055:B5, Sigma-Aldrich) in RPMI with 10% human pooled serum. β−1,3-(D)-glucan (β-glucan) from *Candida albicans* was kindly provided by Professor David Williams (College of Medicine, Johnson City, USA). When indicated, cells were exposed to 5 or 15 μM SC-26196 (Abcam), 5 μM GSK2033 (Tocris), 10 or 100 μM Zileuton (Sigma), 2 or 20 μM ML355 (Cayman Chemical), 10 μM PD146176 (Tocris Bioscience), 0.5 mM Aspirin (Sigma), 1 or 50 μM Celecoxib (Sigma), 5 or 10 μM Itraconazole (Janssen), 30 or 300 nM

FR122047 (Cayman Chemical), 0.01, 0,1 or 1 μM TG4-155 (Focus Bio-molecules), 0.01, 0.1 or 1 μM L798106 (Sigma), 0.1, 1 or 10 μM BGC20-1531 (Cayman Chemical), 0.01, 0.1 or 1 μM Timapiprant (MedChemExpress) or respective vehicle control in combination with BCG for 24 h. In supplementation experiments, monocytes were incubated for 24 h with 0.1, 1 or 5 μM 17-HDHA (Cayman Chemical), 3, 30 or 300 nM 5-R/S-HETE (Cayman Chemical), 0.03, 0.3 or 3 μM 15- HETE (Cayman Chemical), 0.1, 1 or 5 μM 7-HDHA (Cayman Chemical), 1 or 10 nM Leukotriene B$_4$ (Cayman Chemical) or 10 or 100 pM PGE$_2$ (Sigma), 0.1 or 10 nM 12-R/S-HEPE (Cayman Chemical), 0.1 or 10 nM 12-R/S-HETE (Cayman Chemical), 0.1 or 10 nM 14-HDHA (Cayman Chemical). After 24 h, cells were washed once with warm PBS and left to differentiate in RPMI supplemented with 10% pooled human serum for 5 days. The medium was refreshed on day 3 of culture. At day 6, the monocytes-derived macrophages were restimulated with 10 ng/mL LPS for an additional 24 h. All conditions used were tested for cytotoxicity by measuring lactate dehydrogenase in the conditioned media after 24 h incubation according to the manufacturer's instructions (CytoTox96, Promega) (Fig. S1).

## Lipid analysis

Targeted LM and PUFA analysis were carried out using Liquid chromatography coupled to tandem mass spectrometry (LC-MS/MS). Briefly, $1 × 10^6$ monocytes per sample were washed with PBS and lipid extraction and purification were carried out by solid-phase extraction (SPE)[57,58]. LC-MS/MS analysis was carried out on a QTrap 6500 mass spectrometer (Sciex), connected to a Shimadzu Nexera LC30-system including an autosampler and a column oven (Shimadzu) and acquired in Analyst 1.7 (Sciex). The employed column was a Kinetex C18 50 × 2.1 mm, 1.7 μm, protected with a C8 precolumn (Phenomenex). LC-MS/MS peaks were integrated under manual supervision and area corrected to the corresponding internal standard, an in-house established mix of deuterated PUFA (PGE$_2$ d4, Leukotriene B4 d4, 15-HETE d8, DHAd5), using MultiQuant 2.1 (Sciex). All values are a relative quantitation after IS correction; they are displayed as area ratio and were normalized to protein content measured using Pierce TM BCA protein assay kit (Thermo Fisher Scientific) according to the manufacturer's instructions.

## Cytokine quantification

Cytokine production was determined using commercial ELISA kits for IL-1β, IL-6, and TNFα (R&D Systems), following the instructions of the manufacturer and collected in Gen5 3.03 (Biotek).

## RNA isolation and RT-PCR

Total RNA was extracted by TRIzol (Life Technologies), according to the manufacturer's instructions, and quantified using a Nanodrop 2000 UV-visible spectrophotometer. cDNA was synthesized from 450 ng-1 μg RNA by iScript Reverse Transcriptase (Invitrogen). Relative expression was determined using SYBR Green (Invitrogen) on an Applied Biosciences StepOne PLUS qPCR machine (StepOne v2.3 (Thermo Fisher). Fold changes in expression were calculated by the ΔΔCt method using *HPRT1* as an endogenous control. The following primers were used:XX *HPRT1*: 5'-GGATTTGAAATTCCAGACAAGTTT-3', 5'-GCGATGTCAATAGGACTCCAG-3'; *FADS1*: 5'-CCAACTGCTTCCGCAA AGAC-3', 5'-GCTGGTGGTTGTACGGCATA-3'; *FADS2*: 5'-GGATGGCTG CAACATGATTATGG−3', 5'-GCAGAGGCACCCTTTAAGTGG-3'; *SCD*: 5'-TTCCTACCTGCAAGTTCTACACC-3', 5'-CCGAGCTTTGTAAGAGCGGT-3'; *COX1*: 5'- CGCCAGTGAATCCCTGTTGTT-3', 5'-AAGGTGGCATTGA-CAAACTCC-3'; *COX2*: 5' CTGGCGCTCAGCCATACAG-3', 5'-CGCACT-TATACTGGTCAAATCCC-3'.

## Animal experimental model

Animal experiments were conducted in the Unit of Animals for Medical and Scientific purposes of the University General Hospital "Attikon" (Athens, Greece) according to EU Directive 2010/63/EU for animal experiments and to the Greek law 2015/2001, which incorporates the Convention for the Protection of Vertebrate Animals used for Experimental and Other Scientific Purposes of the Council of Europe (code of the facility EL25BIO014, approval number 1853/2015). All experiments were licensed from the Greek veterinary directorate under the protocol number 338087/27-05-2020. 30 C57BL/6JOlaHsd male mice (Hellenic Pasteur Institute) that were 8–10 weeks old were used. Animals were housed in groups of 5 animals per enriched type-II cage on a 12 h light-dark diurnal cycle with room temperature between 21 and 23 °C. Water and food were provided *ad libitum*. DSS was added to the drinking water in a concentration of 3 %. Mice were split into groups through a randomization table. Daily intraperitoneal injections of 1 mg of SC-26196 (Abcam) dissolved in dimethyl sulfoxide (DMSO, Sigma) in a volume of 50 μL or vehicle were performed starting on day 1. Analgesia was achieved with paracetamol suppositories. Animals were monitored for weight changes, stool consistency, and rectal bleeding (DAI score) for 6 days, after which they were sacrificed using an intramuscular injection of 100 mg/kg ketamine and 10 mg/kg xylazine, followed by cervical dislocation. The experiment was performed in duplicate.

## In vitro and in vivo trained immunity models for genetic analysis

Two trained immunity models were performed with a cohort of healthy individuals of Western European descent from the Human Functional Genomics Project. The 300BCG cohort consists of 325 adults from the Netherlands (44% males and 56% females, age range 18–71 years). The study was approved by the local ethics committee CMO region Arnhem-Nijmegen, NL58553.091.16. In *the* in vitro model, adherent monocytes were incubated with 2 μg/mL of β−1,3-(D)-glucan or 5 μg/mL BCG (BCG-Bulgaria, Intervax, Canada), for 24 h at 37 °C in the presence of 10 % pooled human serum. On day 6, cells were re-stimulated for 24 h with LPS 10 ng/mL (Sigma). In the in vivo model, healthy adults were vaccinated with a standard dose of 0.1 mL BCG (BCG-Bulgaria, InterVax, Canada) intradermally, and blood was collected before, 14 and 90 days after vaccination. At each visit, $5 × 10^5$ PBMCs were stimulated with heat-killed *Staphylococcus aureus* ($10^6$ CFU/mL)[59].

## Genetic analysis

DNA samples of individuals (n = 325) were genotyped using the commercially available SNP chip, Infinium Global Screening Array MD v1.0 from Illumina. Genotype information on approximately 4 million single-nucleotide polymorphisms was obtained upon imputation (MAF > 5% and R2 > 0.3 for imputation quality) (PLINK v1.90b6.18). Both genotype and cytokine data of in vitro trained immunity responses was obtained for a total of 267 individuals from the 300BCG cohort. Three samples were excluded due to medication use (of which one was identified as a genetic outlier), and one sample due to onset of type 1 diabetes during the study. The fold change of cytokine production between trained and non-trained cells was used as a parameter of the trained immunity response, followed by a quality check for cytokine distribution and exclusion of genetic outliers. In the in vivo trained immunity model, genotype and cytokine data were obtained for a total of 296 individuals. In addition to the outliers described for the in vitro QTL mapping, 18 individuals who were vaccinated in the evening hours were excluded (due to the known circadian effect on BCG vaccination[60]), resulting in 278 samples. Raw cytokine concentrations were log-transformed and ratios of cytokine production between the visits were taken as the fold change of cytokine production. For both in vitro and in vivo trained immunity models, the fold change of cytokine production was mapped to genotype data using a linear regression model with age and sex as covariates. We used a cutoff of $p < 9.99 × 10^{-3}$ to identify suggestive QTL associations affecting trained immunity responses. R-package Matrix-eQTL was used for cytokine QTL mapping[61] and visualization was performed in R (v.3.3.2).

## Human BCG vaccination for in vivo lipid analysis

Healthy adults were vaccinated with a standard dose of 0.1 mL BCG (BCG-Denmark, SSI) intradermally, and blood was collected before and 28 days after vaccination. The study was approved by the local ethics committee CMO region Arnhem- Nijmegen, NL74082.091.20. At each visit, monocytes were negatively selected from PBMCs (Pan Monocyte Isolation Kit, Miltenyi Biotec) collected in methanol, and stored at −80 °C. Lipid extraction was performed by collecting the supernatant following protein precipitation with MeOH. Samples were dried under a gentle stream of nitrogen and analyzed using LC-MS/MS as previously described. For PUFA quantification, external calibration lines were constructed. All values were normalized to cell numbers.

## Statistics and reproducibility

Data are presented as mean+SD or median and interquartile region when indicated. Analysis and visualization was performed using GraphPad Prism 5.03 and 9.0.1 (San Diego, CA, USA). When indicated, data was analyzed using a two-way ANOVA followed by Sidak's multiple comparisons, Friedman's test followed by Dunn's multiple comparisons test or a Wilcoxon signed-rank test. A $p$-value below 0.05 was considered statistically significant and mentioned in the figures. In every case a cutoff of 5% for Q for a comparison to be considered significant. No data was excluded in vitro monocytes stimulation experiments nor in in vivo mouse experiments. The genetic analysis excluded three samples due to medication use (of which one was identified as a genetic outlier), and one sample due to the onset of type 1 diabetes during the study. In the in vivo lipid analysis, samples below the limit of detection of the technique and statistically significant outliers were excluded. Regarding sample size, for the in vivo mouse experiments sample size was determined using the Power and Sample Size Calculator using the Fisher exact test, with a power of 80% and a type I error of 0.05, with a null hypothesis that 80% of the control group and 20% of the treatment group would develop a clinically significant colitis, defined as DAI score greater than 2. The number of individuals included in the 300BCG study has been chosen based on prior cohort sizes in the Functional Genomics Project (500FG, 200FG) that demonstrated that such size numbers are sufficient for identifying significant factors for immunological assays.

## Reporting summary

Further information on research design is available in the Nature Portfolio Reporting Summary linked to this article.

# Data availability

All data supporting the findings of this study are available within the article and its supplementary materials, including Source Data. Source data are provided with this paper.

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

## Acknowledgements

We thank all volunteers for their participation in this study. Informed consent was obtained from all subjects involved in the study, according to the approval of the ethical committee Arnhem-Nijmegen (2010/104). AVF is supported by the FSE and Fundação para a Ciência e a Tecnologia (FCT, PhD grant PD/BD/135449/2017). JCAB is supported by a grant from the Administrative Department of Science, Technology and Innovation, COLCIENCIAS, in Colombia (Call 756/2016). JDA is supported by The Netherlands Organization for Scientific Research (VENI grant 09150161910024). MGN is supported by an ERC Advanced Grant (#833247) and a Spinoza Grant of the Netherlands Organization for Scientific Research.

## Author contributions

Conceptualization: A.V.F., J.C.A.-B., M.G.N., M.G.; Methodology: A.V.F., J.C.A.-B., J.D.-A., M.G.N., M.G.; Formal Analysis: A.V.F., J.C.A.-B., VM; Investigation: A.V.F., J.C.A.-B., J.D.-A., O.B., G.K., P.A.D., R.J.R., H.N.O.,

E.T., A.Z., S.K., Y.M., V.M., G.R., E.J.G.-B., Writing—Original Draft: A.V.F., J.C.A.-B.; Writing—Review and Editing: all. All authors have read and agreed to the published version of the manuscript.

## Competing interests

MGN is a scientific founder of TTxD. The remaining authors declare no conflict of interest. The funders had no role in the design of the study; in the collection, analyzes, or interpretation of data; in the writing of the manuscript, or in the decision to publish the results.
