## [Peer Review File · Nature Communications]

Fatty acid desaturation and lipoxygenase pathways support trained immunityREVIEWER COMMENTS

Reviewer #1 (Remarks to the Author):

The study by Ferreira, Alarcon-Barreira et al, have investigated the potential role of lipid mediators in the induction of trained immunity (TI) by two well know training agents, the live vaccine BCG and the fungal cell-wall beta-glucan as well as LPS as a tolerizing agent. The authors used human monocytes stimulated or not with BCG/beta-glucan/LPS for 24h and let the cells differentiate in macrophages. By using MS, they evaluated the lipidomic profile of trained monocytes (24h) and macrophages (day 5 post training) and observed that induction of trained immunity with BCG was associated with changes in the production of several lipid mediators from COX and LOX pathways. They further used chemical inhibitors and exogenous lipids to show the effect of those pathways on induction of trained immunity and identified the lipid mediators from the LOX pathways were critical for TI induction. In addition, trained monocytes from healthy volunteers vaccinated with BCG showed enhanced production of LOX mediators.

While the study is very interesting and open new research areas to the field of trained immunity by targeting lipid mediators, it remains limited by a lack of extensive lipidomic analysis to include more important lipid mediators (see comment below). In addition, the differences observed in the experiments using pathways antagonists or exogenous lipids are not striking and would benefit from individual data point graphical representation. Undoubtedly, additional experiments are required to strengthen the conclusions drawn from this study.

Major comments:

1. The strongest association with the lipid mediators in vitro culture system (monocytes/macrophages) was with BCG infection. While the investigators aimed to study trained immunity (memory), it is not clear to this reviewer that they have studied training or priming. Considering BCG is an intracellular bacteria that replicates in phagocytes, it is unclear whether after 24h infection of monocytes with BCG, the bacteria was completely removed from the system. Thus the constant presence of BCG in this culture system would no longer an indication of trained immunity. The authors need to perform CFU assay at different timepoints after wash to demonstrate that BCG is no longer present in this culture system
2. In the figure 1, the authors used LC/MS-MS to identify the effect of TI induction on the production of various lipid mediators. While some important mediators are shown, many are missing from the analysis, such as other prostanoids (PGF₂, PGI₂, PGD₂), cysteinyl leukotrienes (LTC₄, LTD₄, LTE₄), which are known to contribute to macrophage functions. Thus it is difficult to fully understand the effect of TI induction on lipid mediators production as the picture is not complete. Is there a reason why those LMs are not detected/shown?
2. In Figures 3 and 4, the authors used chemical antagonists of the main enzymes involved in lipid mediators' synthesis, as well as exogenous lipids, to evaluate the role of these pathways on TI induction. They found that inhibition of COX1+COX2, but not COX2 only, or of 15-LOX led to increase cytokine production following LPS stimulation. Inversely, inhibition of 5-LOX and 12-LOX led to decreased cytokine production. While this approach has some values, blocking the main enzyme of the pathway leads to a shunting towards another one. Therefore, one approach that could be used by the authors is to selectively inhibit the production of the different metabolites (such as inhibition of mPges1 for PGE₂ production, etc.) or selectively inhibit the receptors of these lipids (e.g. EP₂, 3, 4, and etc) as many selective receptor antagonists are commercially available.
3. I am also concerned that the experimental model might be limited for the use of inhibitors, as the pharmacokinetic of these drugs are unclear and whether they are still present at the time of LPS stimulation, which can itself induce the production of LM that regulate cytokine production. Would RNAi be a better approach? Alternatively, the authors could assess the cytokine production of mature macrophages, trained or not, treated with the different inhibitors or exogenous lipids.
4. In Figure 3, the authors showed that aspirin (COX1/2i) but not celecoxib (COX2i), had an effect on TNF-alpha production. This is surprising as COX2 expression is highly upregulated. Thus, the rationale of using only exogenous PGE₂ in the rest of the study is not clear? The usage of selective inhibitor of COX1 can address this issue. These experiments would significantly benefit by using specific receptor antagonists of prostanoids.
5. In Figure 4, what is the effect of 12-HETE on induction of TI? The authors also show that inhibition of 15-LOX enhanced cytokine production. However, exogenous 15-HETE treatment

enhanced cytokine production. Please elaborate on this point.

7. The inclusion of the human cohort and SNPs in Figure 5A-B is a strength of this manuscript. However, there is no indication of the functional consequences of these SNPs. Is there any effect on the levels of LM in the serum?

Minor comments

1. In Figure 3B, Celecoxib is a selective inhibitor of COX2, not COX1.

2. Why are the levels of 12-HETE and 14-HDHA in Figure 5C reported as "ng/ml/cells" instead of "mg prot"?

3. In Figure S4, it is indicated n=6 but many graphs have only 4 values?

Reviewer #2 (Remarks to the Author):

In this manuscript, Ferreira/ Alarcon-Barrera et al. showed that the phenomenon of trained innate immunity depends on lipid metabolism pathways or various lipid mediators, respectively.

Overall, this is a timely and interesting study with lots of effort invested and quite an amount of data collected. Given that the identification of novel regulators of the first line of defense, the macrophages, is highly important for potential future therapeutics, this study provides relevant data for the field on innate immunity and their role in the defense against pathogens. Also, it adds new insights into the formation of a trained macrophage through metabolic changes.

However, my main concern about this manuscript is that the immunological phenotypes were not developed enough and that functional data are missing entirely. Furthermore, after carefully reading the manuscript, the main message within the data feels more obscured than brought to light.

Major points:

This study certainly needs a schematic presentation/ graphical abstract to describe what was actually discovered here. There are many data but I had problems linking them to a thread I could follow.

What is so special about BCG? What about other pathogens/vaccines regarding trained immunity and the fatty acid metabolism respectively?

Regarding the model: Why stimulate with BCG and then perform an unspecific/TLR4-specific challenge with LPS? What happens if you challenge with BCG or use LPS on both ends? How stable is the phenotype? Were proliferation or apoptosis measured for different stimulations and even more important for different chemical inhibitors? Are there known side/ off-target effects for said inhibitors?

From an immunological point of view, the analysis of the trained immunity phenotype after BCG exposure is very limited. Measuring secreted cytokines in the supernatant is certainly not wrong, but on the other hand far from state-of-the art. A bulk-seq would give us more insights into the generated macrophage phenotype. Single cell techniques like (at least) flow cytometry would be necessary to see what is actually going on in the well. Additionally, studying myeloid cells and claiming a phenotype, some actual functional assays would be needed (e.g. T cell suppression, migration, phagocytosis, ..). Also metabolic analyses might be of interest in the described settings (seahorse?).

Minor points:

I would recommend explaining abbreviations in the figure legends again.

Although I am aware that this would mean a serious re-design, I would strongly recommend using

dot plots instead of columns or even a mix of both, that single data points can be seen and judged, all the more as the error bars are quite huge in most of the figures.

If in the figure legend, it is written n=6 pooled from 2 independent experiments, does this mean n=6 per group, with 2 biological controls and 3 technical replicates (3 stimulated wells/reaction rooms)?

What does "# vs. control, #,*" mean? I assume # stands for "statistical significant vs. the first column (control)?"

Fig.2A: As there are just three data points each and there are no statistical significances, I have to ask why are there no more data points?

Fig.2D: I wonder why the animal experiment was not exploited further. Were there 15 mice per group? Could you add the individual curves at least in the supplementals? Was there any further analysis, for example of LPMC? In line 347, the authors claim "we show that FADS2 inhibition decreased both trained immunity and the severity of colitis in a murine model." Where are the data for trained immunity in this colitis model? Apologies if I missed this in the manuscript.

Fig. 3A: Isn't Celecoxib a COX2-Inhibitor?

Line 306: "Taking advantage of models," I think there is something missing here.

Fig. 5: figure legends are too short here. I assume for A there were sorted monocytes used in vitro and for B PBMC from the patients. The data for A and B are presented in a very confusing manner. What is the order of the SNPs in the heatmaps? It is very difficult to find or search for the same SNPs over the four heatmaps. Also for the cytokine data, the abbreviations should be explained (TT, TC, CC, ...).

Fig.5A: Do the cytokine data belong to the b-glucan or the BCG samples?

Fig. 5B: Do the cytokine data belong to the Day 14 or day 90 samples? Why was *S. aureus* used? Why working with PBMC instead of myeloid cells (macrophages/monocytes)?

Fig 5C: I am not convinced by these data: the placebo always goes up and for 14-HDA it is just because of one "outlier" that this becomes a difference. Can the authors comment on that?

Line 261: What is the connection between the DSS data and LXRs?

Reviewer #3 (Remarks to the Author):

This manuscript provides an insight into the role played by lipid metabolites on the development of trained immunity using the well-characterized models provided by BCG and beta glucan and using LPS-induced training as a control. The authors seemingly identify some lipid metabolite pathways that are involved in the process and continue to confirm them using pharmacological approaches. Finally, the identification of SNPs associated with the training phenotype in genetic regions that cover some of the genes of interest was pursued. A major concern with the manuscript is the interpretation of the data presented. It is clear that BCG, a complex stimulus, induces some of the changes annotated, while the responses to beta glucan and LPS appear quite similar, compared to non stimulated (RPMI) controls. Therefore, it is difficult to conclude that the manuscript identifies a training response to a simple PAMP like beta glucan but rather to a much more complex stimulus represented by BCG. This difference is rarely acknowledged and statistically speaking, the metabolome induced by the treatment with beta-glucan and LPS are quite similar. It is noteworthy the extensive use of terms such as tendency or trends, which are not validated by the statistical analysis of the data.

Another unclear data set presented is related to the experimentation with the DSS-induced

inflammatory model in mice and the role played by FADS2 activity. The justification provided assigns both the role of polyunsaturated fatty acids and inappropriate induction of trained immunity by mycobacteria, which one assumes is not present in the animal facility in which this experimentation was performed, and it was not provided to the mice prior to DSS treatment. This experiment would be meaningful if the mice had been previously treated with BCG and it was observed that in the context of the manuscript's goal, it results in increased disease that could be counteracted by FADS2 inhibition.

Finally, and regarding Figure 5C, the authors categorically say that the production of 12-HETE, 12-HEPE and 14-HDHA were increased in BCG-vaccinated individuals. However, these data need a proper statistical analysis in order to validate this assertion, such as an interaction test that compares the placebo and BCG-vaccinated groups. This is particularly important for the data regarding 12-HETE in which the slopes of the increases in the placebo group seem quite similar to those observed in the BCG set.

Overall, this is a manuscript with potentially interesting data that nevertheless seems to be over/missinterpreted and lacks some critical controls and statistical analysis.

Response to the Reviewer's Concerns

Reviewer #1: The study by Ferreira, Alarcon-Barreira et al, have investigated the potential role of lipid mediators in the induction of trained immunity (TI) by two well know training agents, the live vaccine BCG and the fungal cell-wall beta-glucan as well as LPS as a tolerizing agent. The authors used human monocytes stimulated or not with BCG/beta-glucan/LPS for 24h and let the cells differentiate in macrophages. By using MS, they evaluated the lipidomic profile of trained monocytes (24h) and macrophages (day 5 post training) and observed that induction of trained immunity with BCG was associated with changes in the production of several lipid mediators from COX and LOX pathways. They further used chemical inhibitors and exogenous lipids to show the effect of those pathways on induction of trained immunity and identified the lipid mediators from the LOX pathways were critical for TI induction. In addition, trained monocytes from healthy volunteers vaccinated with BCG showed enhanced production of LOX mediators. While the study is very interesting and open new research areas to the field of trained immunity by targeting lipid mediators, it remains limited by a lack of extensive lipidomic analysis to include more important lipid mediators (see comment below). In addition, the differences observed in the experiments using pathways antagonists or exogenous lipids are not striking and would benefit from individual data point graphical representation. Undoubtedly, additional experiments are required to strengthen the conclusions drawn from this study.

Major comments:

1. The strongest association with the lipid mediators in vitro culture system (monocytes/macrophages) was with BCG infection. While the investigators aimed to study trained immunity (memory), it is not clear to this reviewer that they have studied training or priming. Considering BCG is an intracellular bacteria that replicates in phagocytes, it is unclear whether after 24h infection of monocytes with BCG, the bacteria was completely removed from the system. Thus the constant presence of BCG in this culture system would no longer an indication of trained immunity. The authors need to

perform CFU assay at different timepoints after wash to demonstrate that BCG is no longer present in this culture system

Response:

We thank the reviewer for this pertinent question. We performed a CFU assay as suggested and we observed that BCG is still present at day 1 and day 6 of culture (Extra Figure 1). Specifically, of the average 5756 BCG CFU we added to 0.1 million cells, we obtained on average 518 intracellular BCG CFUs after 24h (D1) and 988 intracellular BCG CFU after the 6 days (D6) of culture.

Extra Figure 1. Quantification of BCG during the *in vitro* trained immunity model. (A) Monocytes were infected with 5 μ g/mL BCG and CFUs were determined at in infection media at day 0 (D0), intracellularly after 24h (D1) and 6 days of culture (D6) (n=6 pooled from 2 independent experiments). (B) Monocytes were infected with 5 μ g/mL BCG and percentage of BCG+ cells was determined after 24h (D1) and 6 days of culture (D6) (n=6, mean+SD, Wilcoxon signed-rank test).

Although BCG is present in the culture at day 6, we do not observe striking transcriptional changes when compared to RPMI control cells at the same timepoint (Extra Figure 2 left panel). In contrast, in the LPS tolerance model (PMID: 27863248) we observed a higher number of differentially expressed genes (Extra Figure 2 right panel).

Extra Figure 2 Enhanced volcano plot of differentially expressed genes in macrophages at day 6 of culture after BCG or LPS stimulation. Monocytes were exposed to BCG, LPS or control conditions for 24h, washed and then left to differentiate for an additional 5 days (n=3).

We have also observed that IL-6 production is increased in monocytes after 24h of BCG exposure. However, at day 6 after BCG exposure, the levels of this cytokine resolves to concentrations similar to control cells (Extra Figure 3). The transcriptional and cytokine data presented here do not support a continuous cell activation status change derived from the presence of BCG in the culture.

Extra Figure 3 IL-6 production at day 1, day 3, day 6 and day 7 of the in vitro trained immunity protocol using BCG as a stimulus for the first 24h and LPS as restimulation at day 6 of culture. (n=6, mean+SD) dashed line represents the lower limit of detection of the assay.

2. In the figure 1, the authors used LC/MS-MS to identify the effect of TI induction on the production of various lipid mediators. While some important mediators are shown, many are missing from the analysis, such as other prostanoids (PGF₂, PGI₂, PGD₂), cysteinyl leukotrienes (LTC₄, LTD₄, LTE₄), which are known to contribute to macrophage functions. Thus it is difficult to fully understand the effect of TI induction on lipid mediators production as the picture is not complete. Is there a reason why those LMs are not detected/shown?

Response:

The applied assay measured all of these substances, however we found a selective production of PGE₂ with the other compounds being below our detection limit of on average 0.01 ng/mL.

3. In Figures 3 and 4, the authors used chemical antagonists of the main enzymes involved in lipid mediators' synthesis, as well as exogenous lipids, to evaluate the role of these pathways on TI induction. They found that inhibition of COX1+COX2, but not COX2 only, or of 15-LOX led to increase cytokine production following LPS stimulation. Inversely, inhibition of 5-LOX and 12-LOX led to decreased cytokine production. While this approach has some values, blocking the main enzyme of the pathway leads to a shunting towards another one. Therefore, one approach that could be used by the authors is to selectively inhibit the production of the different metabolites (such as inhibition of mPges1 for PGE₂ production, etc.) or selectively inhibit the receptors of these lipids (e.g. EP₂, 3, 4, and etc) as many selective receptor antagonists are commercially available.

Response:

As mentioned by the reviewer, inhibiting the main enzyme of a pathway may lead to the increase of a parallel pathway that competes for the same substrate. In this study, we observed that LOX derived metabolites were enhanced by BCG. Specifically, 12-LOX metabolites were enriched upon 24h *in vitro* stimulation and pharmacological inhibition

of 5-LOX or 12-LOX decreased BCG-induced responsiveness. However, it is quite challenging to comprehensively inhibit the receptors of LOX products due to their promiscuity (PMID: 32670289; 18834304; 31187940) and the lack of proper agonists. Hence, we focused on the selective inhibition of EP2, EP3, EP4 and DP2 as suggested by the reviewer. The inhibition of these receptors did not modify the BCG-enhanced IL-6 production upon LPS stimulation (modified Figure S1A, new Figure S5A). In addition, we also extended our QTL analysis of the *in vivo* 300BCG cohort (Figure 5B) to include SNPs in the vicinity ($\pm 250\text{kb}$) of different receptors for lipid mediators. We focused on prostaglandin and thromboxane receptors *PTGDR2*, *PTGER1-3*, *PTGIR*, *TBXA2R*, *LTB4R* (new Figure S5B) and we have identified SNPs that were suggestively associated ($p < 9.99 \times 10^{-3}$) with changes in $\text{TNF}\alpha$, IL-6 and IL-1 β upon *ex vivo* restimulation 14 days and 90 days after vaccination. Although we did not validate these findings in our *in vitro* model, we cannot exclude that prostaglandin signalling may play a role in the context of *in vivo* trained immunity. We also performed an identical QTL analysis focusing on oxylipins receptors *CMKLR1*, *FPR2*, *GPR18*, *GPR31*, *GPR32*, *GPR37*, *LGR6*, *OXER1* (new Figure S6) which encode for receptors of different lipid mediators and their precursors (PMID: 32670289; 18834304; 31187940)). We have identified SNPs within 250kb of these genes of interest that were suggestively associated ($p < 9.99 \times 10^{-3}$) with changes in $\text{TNF}\alpha$, IL-6 and IL-1 β in the BCG vaccination cohort, possibly supporting our *in vitro* findings, where we have observed that LOX-derived mediators promote increased cell responsiveness. These new findings were added in the new version of the manuscript (page 12, lines 277-279; page 14, lines 324-329; page 17, lines 391/402-404).

4. I am also concerned that the experimental model might be limited for the use of inhibitors, as the pharmacokinetic of these drugs are unclear and whether they are still present at the time of LPS stimulation, which can itself induce the production of LM that regulate cytokine production. Would RNAi be a better approach? Alternatively, the

authors could assess the cytokine production of mature macrophages, trained or not, treated with the different inhibitors or exogenous lipids.

Response:

As suggested by the reviewer, we assessed cytokine production of mature trained macrophages. We observed that 12-LOX inhibition by ML355 decreased BCG-enhanced IL-6 production. Since, we also validated the enrichment in 12-LOX products in monocytes isolated from BCG vaccinated individuals we focused on the involvement of 12-LOX in this assay. Thus, to determine if ML355 could play a direct role in LPS stimulation, we stimulated macrophages with LPS in combination with ML355 (Extra Figure 2). BCG-enhanced IL-6 production did not change upon treatment with 2 μ M ML355 in combination with LPS when compared to LPS alone. The higher dose of 20 μ M ML355 did significantly decrease IL-6 production. The previously seen reduction of IL-6 secretion in cells treated with BCG+ ML355 (Figure 4A middle panel) was observed using 20 μ M ML355, however these cells undergo 2 washes/media changes and are left to differentiate for 5 days after ML355 exposure. We assume that during that time period the concentration of ML355 decreases and at the time of LPS stimulation at day 6 of culture, ML355 must be lower than 20 μ M, while we have observed that 2 μ M ML355 has no effect on IL-6 production triggered by LPS.

Extra Figure 2. 12LOX inhibition concomitant to LPS stimulation decreases IL-6 production enhanced by BCG trained immunity. Production of IL-6 by BCG-trained macrophages incubated with 2 μ M or 20 μ M ML355 and LPS for 24h at the 6th day of culture. (n=9, pooled from 3 independent experiments) Mean+SD, *p<0.05 Friedman test, Dunn's multiple comparisons test.

5. In Figure 3, the authors showed that aspirin (COX1/2i) but not celecoxib (COX2i), had an effect on TNF-alpha production. This is surprising as COX2 expression is highly upregulated. Thus, the rationale of using only exogenous PGE2 in the rest of the study is not clear? The usage of selective inhibitor of COX1 can address this issue. These experiments would significantly benefit by using specific receptor antagonists of prostanoids.

Response:

We have assessed the role of COX1 in BCG-induced cytokine production by inhibiting COX1 pharmacologically with FR122047. We did not observe any changes in TNF α or IL-6 production in the presence of this inhibitor (new Figure 3 right top panel). As also suggested and discussed in point 3, we inhibited pharmacologically EP2, EP3, EP4 and DP2. The inhibition of these receptors did not modify the BCG-enhanced IL-6 production (modified Figure S1A, new Figure S5A). These results were described and discussed in page 12 lines 277-279; page 17 line 391 in the new version of the manuscript.

6. In Figure 4, what is the effect of 12-HETE on induction of TI?

Response:

We have now tested not only 12-HETE but also 14-HDHA, both products of 12-LOX activity, in supplementation experiments. We have observed that 12-HETE, similarly to 12-HEPE, enhanced TNF α and IL-6 production upon LPS stimulation, while 14-HDHA did not (new Figure 4D). 12-HETE and 14-HDHA were increased in BCG exposed monocytes (Figure 1), however their effect on responsiveness does not appear to be similar. These new results were included in page 12/13, lines 290-295, in the new version of the manuscript.

7. The authors also show that inhibition of 15-LOX enhanced cytokine production. However, exogenous 15-HETE treatment enhanced cytokine production. Please elaborate on this point.

Response:

A new discussion on the role of 15-LOX and 15-HETE has been added in the new version of the manuscript (page 13, lines 295-300).

8. The inclusion of the human cohort and SNPs in Figure 5A-B is a strength of this manuscript. However, there is no indication of the functional consequences of these SNPs. Is there any effect on the levels of LM in the serum?

Response:

As suggested by the reviewer, we assessed the serum LM levels in a subset of the human cohort represented in Figure 5B. We analysed serum samples from 20 sex and age matched individuals: 10 with the CC and 10 with the TT genotype of the rs955461 SNP. The rs955461 SNP is on the vicinity (\pm 250kb) of 12LOX and was suggestively associated ($p= 0,000318$) with the fold change of *ex vivo* IL-6 production 14 days after BCG vaccination (Figure 5B, upper boxplot). However, we did not observe an association between this SNP and the levels of mediators measured (eg. AA, DHA, 12-HETE, 14-HDHA) nor where the levels of these metabolites associated with the fold change of IL-6 production. It is important to note that we only had access to serum samples, which do not necessarily reflect intracellular amounts. Additionally, these samples were previously thawed which may affect their quality.

Minor comments

1. In Figure 3B, Celecoxib is a selective inhibitor of COX2, not COX1.

Response:

This error has been corrected in the new version of the manuscript.

2. Why are the levels of 12-HETE and 14-HDHA in Figure 5C reported as “ng/ml/cells” instead of “mg prot”?

Response:

In order to better highlight the effect of BCG vaccination, Figure 5C has now been updated in the new version of the manuscript. After normalization to protein levels, we now present fold changes between visit 2 and visit 1.

3. In Figure S4, it is indicated n=6 but many graphs have only 4 values?

Response:

The measurements present in Figure S4 (Figure S7 in the new version of the manuscript) were performed in 6 biological replicates. Due to the high variability of the samples, for some of the lipid mediators there were replicates that were below the limit of detection and quantitation of the methodology used. Additionally, one participant of each group was an outlier and thus removed. In those cases, the number of replicates reported are less than 6. This is now indicated in the new Figure legend (Page 23, Line 549,550)

Reviewer #2: *In this manuscript, Ferreira/ Alarcon-Barrera et al. showed that the phenomenon of trained innate immunity depends on lipid metabolism pathways or various lipid mediators, respectively.*

Overall, this is a timely and interesting study with lots of effort invested and quite an amount of data collected. Given that the identification of novel regulators of the first line of defense, the macrophages, is highly important for potential future therapeutics, this study provides relevant data for the field on innate immunity and their role in the defense against pathogens. Also, it adds new insights into the formation of a trained macrophage through metabolic changes.

However, my main concern about this manuscript is that the immunological phenotypes were not developed enough and that functional data are missing entirely. Furthermore, after carefully reading the manuscript, the main message within the data feels more obscured than brought to light.

Major points:

1. This study certainly needs a schematic presentation/ graphical abstract to describe what was actually discovered here. There are many data but I had problems linking them to a thread I could follow.

Response:

We thank the reviewer for the comments and the suggestions. We have now included a graphical abstract summarizing the findings of the present manuscript.

2. What is so special about BCG? What about other pathogens/vaccines regarding trained immunity and the fatty acid metabolism respectively?

Response:

Indeed, some vaccines other than BCG induce cross-protection against infections outside the target disease (PMID: 32461674), particularly vaccines that contain live attenuated microorganisms. The measles-containing vaccines (such as measles,

mumps, and rubella [MMR]), and the oral polio vaccine (OPV) have these properties, as they improve mortality in children beyond the protection against their respective target diseases (PMID: 27737834). In the present manuscript, we focused on BCG since (1) it is one of the most widely used vaccines in the world, and thus is widely studied epidemiologically (reviewed in PMID: 20716675) and in randomized trials (PMID: 29996082; 15652667); (2) the mechanisms underlying BCG trained immunity have been the focus of research for the past years (page 3, lines 41- 46). Several studies have investigated the regulation of BCG trained immunity in different models, with *in vitro* approaches, resorting to mouse studies, and using human vaccination models (for example PMID: 27926861, 25258083, 27866838, 29328912, 29328910, 32320649, 32544459, 29324233). Pathways like glycolysis, TCA, and oxidative phosphorylation (PMID: 27926861, 32320649) have been explored in BCG and β -glucan trained immunity, however lipid pathways were not the focus of these studies. Some findings have pointed to the role of lipids in trained immunity: cholesterol biosynthesis was shown to be increased in hematopoietic stem and progenitor cells of β -glucan trained mice (PMID: 29328910), and the metabolite mevalonate, involved in the cholesterol synthesis pathway amplified trained immunity in human monocytes (PMID: 29328908) (page 3, lines 50-58). Taken together, we considered that exploring the BCG-trained immunity program would present the most potential in terms of translational consequences, and we aimed to address the previously underexplored field of lipid mediators in the metabolic regulation of trained immunity.

3. Regarding the model: Why stimulate with BCG and then perform an unspecific/TLR4-specific challenge with LPS? What happens if you challenge with BCG or use LPS on both ends? How stable is the phenotype? Were proliferation or apoptosis measured for different stimulations and even more important for different chemical inhibitors? Are there known side/ off-target effects for said inhibitors?

Response:

In the present study we used the *in vitro* trained immunity protocol (PMID: 33718890), which has been extensively used in previous studies (for example PMID: PMID: 27926861, 25258083, 27866838). We opted to use LPS as secondary challenge since that is the most commonly used secondary challenge in the literature of trained immunity and thus more easily comparable to previous reports. If we were to use BCG as a primary and secondary challenge, the results we would observe could potentially be restricted to BCG.

As the reviewer suggested, we performed a tolerance model (PMID: 33293712, 27863248), where cells are stimulated with LPS for the first 24h culture followed by another 24h at day 6 of culture. We inhibited FASD2, 5-LOX, 12-LOX or 15-LOX and did not observe any changes in the decreased production of TNF α and IL-6 (Extra Figure 4). However, we have not included these results in the new version of the manuscript since we would like to focus on trained immunity and not tolerance responses.

Extra Figure 4. Inhibition of FADS2, 5-, 12-, or 15-LOX did not alter IL-6 or TNF α production in an *in vitro* LPS tolerance model. IL-6 and TNF α production of monocytes previously exposed to LPS and SC26196, Zileuton, ML355 or PD146176 for 24h, left to differentiate for 5 days and restimulated with LPS for another 24h (n=3-9, pooled from 1-3 independent experiments).

Regarding cell proliferation, in a previous report of our laboratory in which the same *in vitro* model was used, cell number at day 6 of culture was not different between control and BCG exposed cells (PMID: 27733422). For the assessment of cell death, we measured the presence of lactate dehydrogenase (LDH) in the conditioned media to determine the cytotoxicity of the inhibitors used (modified Figure S1).

Regarding known side/ off-target effects of the inhibitors used that modulated BCG responsiveness:

- The FADS2 inhibitor SC-26196 has been reported in HepG2 cells to have no cross-reactivity towards $\Delta 5$ and $\Delta 9$ desaturases (PMID: 9765335, 20086206);
- The LXR antagonist GSK2033 was shown to perform as a LXR antagonist (10 μ M HepG2 cells for 24h). However, GSK2033 was shown to be promiscuous by

- targeting a number of other nuclear receptors, but the published report does not specify the concentrations of GSK2033 tested (PMID: 20345102, 27680310);
- Aspirin covalently modifies the COX enzymes through acetylation of Ser530 near its active site, which prevents proper binding of the native substrate and thus leads to its irreversible inhibition. However, aspirin is 10–100 times more potent against COX-1 than against COX-2 (PMID: 8265610, 25514511)
 - The 5-LOX inhibitor zileuton is used in the prophylaxis and treatment of chronic asthma. However, zileuton can induce liver toxicity, which appears to be unrelated to the inhibition of 5-LOX (PMID: 11509739). Zileuton has been shown to suppress PGE₂ synthesis, by interfering at the level of arachidonic acid release (PMID: 20880396). Other 5-LOX inhibitors (AA-861, BWA4C, CJ-13,610, C06) have also been tested and affected PGE₂ metabolism (PMID: 35126121).
 - The 12-LOX inhibitor ML355 has a favourable selectivity with 15-LOX-1/12-LOX ratio of 29 and the IC₅₀ for 15-LOX-2 and 5-LOX are over 100µM (PMID: 25506969, 24393039).
 - PD146176 is a selective 15-LOX inhibitor (PMID: 9105693, 9543090).

It is indeed important to bear in mind that there is no drug free of possible off-target effects (as discussed in Page 13, Line 298) However, the use of pharmacological agents over resorting to genetic tools might be a step closer to the translation of our findings to a human setting.

4. From an immunological point of view, the analysis of the trained immunity phenotype after BCG exposure is very limited. Measuring secreted cytokines in the supernatant is certainly not wrong, but on the other hand far from state-of-the art. A bulk-seq would give us more insights into the generated macrophage phenotype. Single cell techniques like (at least) flow cytometry would be necessary to see what is actually going on in the well. Additionally, studying myeloid cells and claiming a phenotype, some actual functional

assays would be needed (e.g. T cell suppression, migration, phagocytosis). Also metabolic analyses might be of interest in the described settings (seahorse?)

Response:

As rightfully mentioned by the reviewer, trained immunity is not limited to cytokine secretion, however it is the standard readout used in the field. In the last years, our group and others have published different studies on trained immunity that include single cell techniques, such as single cell sequencing and flow cytometry (PMID: 35133977; 37155329; 32544459). Additionally, other techniques such as Seahorse (PMID: 32320649) and phagocytic and killing activity (PMID: 31484076, 33207187) have also been used in other reports. In the present study we focus instead on the analysis of lipid mediators upon trained immunity induced by BCG.

Minor points:

1. I would recommend explaining abbreviations in the figure legends again.

Response:

In the new version of the manuscript, an explanation for abbreviations were included in the figure legends.

2. Although I am aware that this would mean a serious re-design, I would strongly recommend using dot plots instead of columns or even a mix of both, that single data points can be seen and judged, all the more as the error bars are quite huge in most of the figures.

Response:

We agree with the reviewer that this would improve the readability of the presentation and interpretation of the data in our manuscript. Accordingly, single data points can now be seen in the new version of the manuscript.

3. If in the figure legend, it is written n=6 pooled from 2 independent experiments, does this mean n=6 per group, with 2 biological controls and 3 technical replicates (3 stimulated wells/reaction rooms)?

Response:

N=6 pooled from 2 independent experiments means 6 biologicals replicates (6 independent human monocyte donors), divided between 2 independent experiments (2 different dates, different reagent preparations). This has been clarified in the figure legends of the new version of the manuscript.

*4. What does “# vs. control, #, **” mean? I assume # stands for “statistical significant vs. the first column (control)?*

Response:

The assumption of the reviewer is correct. To increase readability, this has been clarified in the figure legends of the new version of the manuscript.

5. Fig.2A: As there are just three data points each and there are no statistical significances, I have to ask why are there no more data points?

Response:

As suggested by the reviewer, we increased the number of biological replicates (Figure 2A and Figure 3B).

6. Fig.2D: I wonder why the animal experiment was not exploited further. Were there 15 mice per group? Could you add the individual curves at least in the supplementals? Was there any further analysis, for example of LPMC? In line 347, the authors claim “we show that FADS2 inhibition decreased both trained immunity and the severity of colitis in a murine model.” Where are the data for trained immunity in this colitis model? Apologies if I missed this in the manuscript.

Response:

As suggested by the reviewer, we have added the individual curves of the 15 mice per group (new Figure S4C). Unfortunately, it was not possible to perform any further analysis outside of documentation of mice weight, which we have also added in supplementation (new Figure S4D). In order to decrease the focus on this experiment, Fig 2D in the previous version of the manuscript was moved to Figure S4A, B in the new version of the manuscript.

We understand how the sentence in line 347 might be misconstrued. We meant that FADS2 inhibition decreases trained immunity in our *in vitro* model (Figure 2C) and decreases the severity of colitis in the murine model (new Figure S4). This sentence has been restructured in the new version of the manuscript (page 15, line 364/365).

7. Fig. 3A: Isn't Celecoxib a COX2-Inhibitor?

Response:

This mistake has been corrected in the new version of the manuscript.

8. Line 306: "Taking advantage of models," I think there is something missing here.

Response:

This sentence has been corrected in the new version of the manuscript.

9. Fig. 5: figure legends are too short here. I assume for A there were sorted monocytes used in vitro and for B PBMC from the patients. The data for A and B are presented in a very confusing manner. What is the order of the SNPs in the heatmaps? It is very difficult to find or search for the same SNPs over the four heatmaps. Also for the cytokine data, the abbreviations should be explained (TT, TC, CC, ...).

Response:

Figure 5A pertains to adherent monocytes isolated by density gradient (page 7, line 162), while in Figure 5B indeed PBMCs isolated from BCG vaccinated (prior, 14 days and 90 days after) healthy individuals (page 8, line 167).

Regarding the order of the SNPs in the heatmaps, we opted to maintain the order of the genes consistent between heatmaps, as different heatmaps do not necessarily show the same SNPs. The SNPs here presented are the top hits (lowest p -value) in each cytokine-training stimulus pair. If there are two SNPs that are not in linkage disequilibrium (LD), we represented both in the heatmap. It is important to note that the mentioned SNPs are not necessarily causal to the phenotype studied, they could be in LD with the causal SNP and still be the most strongly associated (lowest p -value) with the trait (in this case change in cytokine production).

As suggested by the reviewer, to increase readability we have altered the figure and extended the legend to include an explanation of abbreviations.

10. Fig.5A: Do the cytokine data belong to the β -glucan or the BCG samples?

Response:

In Figure 5A, we can observe a heatmap which corresponds to the fold change of TNF and IL-6 upon β -glucan-induced trained immunity (Figure 5A top left panel) and BCG-induced trained immunity (Figure 5A bottom left panel). The boxplots on the right correspond to the SNPs in bold on the heatmaps, which are the one with the lowest p -values. The boxplot of *FADS2* rs9326418 (Figure 5A top right) is in bold in the β -glucan heatmap and the boxplot of *LOX12* rs12232535 (Figure 5A bottom right) is in bold in the BCG heatmap. To increase readability, we have modified this figure in the new version of the manuscript.

*11. Fig. 5B: Do the cytokine data belong to the Day 14 or day 90 samples? Why was *S. aureus* used? Why working with PBMC instead of myeloid cells (macrophages/monocytes)?*

Response:

In Figure 5B we can observe a heatmap which corresponds to the fold change of TNF, IL-6 and IL-1 β between Day 14 and Day 0 (Figure 5B top left panel) and a heatmap that

pertains to Day 90 and Day 0 (Figure 5B bottom left panel). The boxplots on the right correspond to the SNPs in bold on the heatmaps, which are the one with the lowest *p*-values. The boxplot of *LOX12* rs955461 (Figure 5B top right) is in bold in the Day 14 heatmap and the boxplot of *FADS2* rs11600019 (Figure 5B bottom right) is in bold in the Day 90 heatmap. To increase readability, we have modified this figure in the new version of the manuscript.

For the analysis present in Figure 5B, we used a previously established cohort (PMID: 32692728, 36094960). In this cohort *S. aureus* was used as a proxy for heterologous stimulus. The cohort used PBMCs instead of monocytes not only to assess the concentrations of the monocytes derived cytokines, such as IL-6, TNF and IL-1 β , but also to evaluate the potential of the BCG vaccine to enhance T cell-derived cytokines upon a heterologous stimulus. However, in the present study we focused on myeloid-derived cytokines.

12. Fig 5C: I am not convinced by these data: the placebo always goes up and for 14-HDA it is just because of one "outlier" that this becomes a difference. Can the authors comment on that?

Response:

This is also a fair point raised by the reviewer. The possible seasonal effect is indeed one reason why we included the placebo group in the study. We have adjusted Figure 5C in the new version of the manuscript to show a fold change between V2 and V1 instead of absolute quantification, in order to better illustrate the effect of BCG vaccination. There is no statistical difference between visits in the placebo group. On the other hand, the BCG vaccinated group showed a statistically significant increase of 12-HEPE and 14-HDHA while 12-HETE approached significance.

13. Line 261: What is the connection between the DSS data and LXRs?

Response:

LXR is a transcription factor that regulates fatty acid homeostasis. Particularly, LXR controls the transcription of *FADS2* (PMID: 25838428; 31658997; 28041958). In the present manuscript, we observed that LXR inhibition (as also reported in PMID: 35682840) decreases BCG-enhanced IL-6 and TNF production (new Figure 2D) as does *FADS2* inhibition (new Figure 2C). Additionally, we report that *FADS2* inhibition in a DSS mouse colitis model decreases disease score (new Figure S4). We have added a sentence to clarify this connection in the new version of the manuscript (Page 11, Line 263-265)

Reviewer #3: *This manuscript provides an insight into the role played by lipid metabolites on the development of trained immunity using the well-characterized models provided by BCG and beta glucan and using LPS-induced training as a control. The authors seemingly identify some lipid metabolite pathways that are involved in the process and continue to confirm them using pharmacological approaches. Finally, the identification of SNPs associated with the training phenotype in genetic regions that cover some of the genes of interest was pursued.*

1. TA major concern with the manuscript is the interpretation of the data presented. It is clear that BCG, a complex stimulus, induces some of the changes annotated, while the responses to beta glucan and LPS appear quite similar, compared to non stimulated (RPMI) controls. Therefore, it is difficult to conclude that the manuscript identifies a training response to a simple PAMP like beta glucan but rather to a much more complex stimulus represented by BCG. This difference is rarely acknowledged and statistically speaking, the metabolome induced by the treatment with beta-glucan and LPS are quite similar. It is noteworthy the extensive use of terms such as tendency or trends, which are not validated by the statistical analysis of the data.

Response:

We thank the reviewer for the comments and the suggestions. In the present manuscript, we focus on the lipid changes induced by BCG, since indeed this mycobacterium induces a more robust acute modulation than the one observed by the PAMP derived from *C. albicans* β -glucan. Thus, we suggest that the lipid mediators studied here are of more relevance to BCG-induced trained immunity than to β -glucan-induced trained immunity. At the concentrations tested in this manuscript BCG and β -glucan induce trained immunity while LPS induces immune tolerance. However, although BCG and β -glucan both trigger increased responsiveness, the mechanisms that underline this potentiation are not necessarily common between the two inducers (PMID: 27733422).

Our data demonstrates that the monocytic response to the trained immunity inducer β -glucan is not the same as the one to the tolerogenic LPS. Particularly at day 6 of culture, cells previously stimulated with LPS stimulated showed increased levels of linoleic acid (LA), alpha-linoleic acid (ALA) (Figure S2), PGE2 and 5-HETE (Figure S3) and decreased amounts different mediators, such as LTB4, 17-HDHA, 15-HEPE, 5-HEPE and 18-HEPE (Figure S3), when compared to controls or β -glucan trained cells. However, these differences were not tested statistically. The number of biological replicates for LPS stimulation is lower than the other treatments since this condition was not the focus of this particular study. An additional short description of the results derived from the tolerance model was added in the new version of the manuscript (Page 10, Line 232/233)

2. Another unclear data set presented is related to the experimentation with the DSS-induced inflammatory model in mice and the role played by FADS2 activity. The justification provided assigns both the role of polyunsaturated fatty acids and inappropriate induction of trained immunity by mycobacteria, which one assumes is not present in the animal facility in which this experimentation was performed, and it was not provided to the mice prior to DSS treatment. This experiment would be meaningful if the mice had been previously treated with BCG and it was observed that in the context of the manuscript's goal, it results in increased disease that could be counteracted by FADS2 inhibition.

Response:

We understand the reviewer's comment. To decrease the focus of this experiment, Figure 2D was moved to new Figure S4. However, we consider that it is relevant to not remove this model from the study since it has implications for the field of inflammatory bowel disease research.

3. Finally, and regarding Figure 5C, the authors categorically say that the production of 12-HETE, 12-HEPE and 14-HDHA were increased in BCG-vaccinated individuals. However, these data need a proper statistical analysis in order to validate this assertion, such as an interaction test that compares the placebo and BCG-vaccinated groups. This is particularly important for the data regarding 12-HETE in which the slopes of the increases in the placebo group seem quite similar to those observed in the BCG set.

Response:

As suggested by the reviewer, we used regression analysis to investigate any interaction between the treatment outcome and sex, age, or BMI. There were no significant interactions with sex and BMI; however interaction between treatment and age was present with significant p-values; at p-value of 0.005 for BCG treatment and age in 12-HETE, at 0.004 and 0.023 for BCG and Placebo in 12-HEPE, and at 0.0025 for BSG in 14-HDHA. This effect persisted after removing outliers with the highest values in each group, which correspond to the oldest subjects in the BCG as well as in the placebo group. The effect of age was positive in all instances, i.e. higher concentration of 12-HETE, 12-HEPE and 14-HDHA were measured in older individuals. However, we do not anticipate that this interaction would influence the findings since the ages in placebo and BCG groups were matched (Placebo: 30,83 years \pm 11,2; BCG: 29,3 years \pm 10,5, mean \pm SD).

REVIEWERS' COMMENTS

Reviewer #1 (Remarks to the Author):

The authors experimentally addressed all my concerns and I have no further comment.

Reviewer #2 (Remarks to the Author):

I thank the authors for answering all my questions and for improving the manuscript.

Major points:

1. This study certainly needs a schematic presentation/ graphical abstract to describe what was actually discovered here. There are many data but I had problems linking them to a thread I could follow.

Response:

We thank the reviewer for the comments and the suggestions. We have now included a graphical abstract summarizing the findings of the present manuscript.

Thank you.

2. What is so special about BCG? What about other pathogens/vaccines regarding trained immunity and the fatty acid metabolism respectively?

Response:

Indeed, [...] Taken together, we considered that exploring the BCG-trained immunity program would present the most potential in terms of translational consequences, and we aimed to address the previously underexplored field of lipid mediators in the metabolic regulation of trained immunity.

Thank you.

3. Regarding the model: Why stimulate with BCG and then perform an unspecific/TLR4-specific challenge with LPS? What happens if you challenge with BCG or use LPS on both ends? How stable is the phenotype? Were proliferation or apoptosis measured for different stimulations and even more important for different chemical inhibitors? Are there known side/ off-target effects for said inhibitors?

Response:

In the present study we used the in vitro trained immunity protocol (PMID: 33718890), which has been extensively used in previous studies (for example PMID: PMID: 27926861, 25258083, 27866838). We opted to use LPS as secondary challenge since that is the most commonly used secondary challenge in the literature of trained immunity and thus more easily comparable to previous reports. If we were to use BCG as a primary and secondary challenge, the results we would observe could potentially be restricted to BCG.

Thank you.

As the reviewer suggested, we performed a tolerance model (PMID: 33293712, 27863248), where cells are stimulated with LPS for the first 24h culture followed by another 24h at day 6 of culture. We inhibited FASD2, 5-LOX, 12-LOX or 15-LOX and did not observe any changes in the decreased production of TNF α and IL-6 (Extra Figure 4).

However, we have not included these results in the new version of the manuscript since we would like to focus on trained immunity and not tolerance responses.

Thank you.

Regarding cell proliferation, in a previous report of our laboratory in which the same in vitro model was used, cell number at day 6 of culture was not different between control and BCG exposed cells (PMID: 27733422). For the assessment of cell death, we measured the presence of lactate

dehydrogenase (LDH) in the conditioned media to determine the cytotoxicity of the inhibitors used (modified Figure S1).

Thank you, these data are an important add-on.

Regarding known side/ off-target effects of the inhibitors used that modulated BCG responsiveness:[...]

It is indeed important to bear in mind that there is no drug free of possible off-target effects (as discussed in Page 13, Line 298) However, the use of pharmacological agents over resorting to genetic tools might be a step closer to the translation of our findings to a human setting.

Thank you.

4. From an immunological point of view, the analysis of the trained immunity phenotype after BCG exposure is very limited. Measuring secreted cytokines in the supernatant is certainly not wrong, but on the other hand far from state-of-the art. A bulk-seq would give us more insights into the generated macrophage phenotype. Single cell techniques like (at least) flow cytometry would be necessary to see what is actually going on in the well.

Additionally, studying myeloid cells and claiming a phenotype, some actual functional assays would be needed (e.g. T cell suppression, migration, phagocytosis). Also metabolic analyses might be of interest in the described settings (seahorse?)

Response:

As rightfully mentioned by the reviewer, trained immunity is not limited to cytokine secretion, however it is the standard readout used in the field. In the last years, our group and others have published different studies on trained immunity that include single cell techniques, such as single cell sequencing and flow cytometry (PMID: 35133977; 37155329; 32544459). Additionally, other techniques such as Seahorse (PMID: 32320649) and phagocytic and killing activity (PMID: 31484076, 33207187) have also been used in other reports. In the present study we focus instead on the analysis of lipid mediators upon trained immunity induced by BCG.

Thank you.

Minor points:

1. I would recommend explaining abbreviations in the figure legends again.

Response:

In the new version of the manuscript, an explanation for abbreviations were included in the figure legends.

Thank you.

2. Although I am aware that this would mean a serious re-design, I would strongly recommend using dot plots instead of columns or even a mix of both, that single data points can be seen and judged, all the more as the error bars are quite huge in most of the figures.

Response:

We agree with the reviewer that this would improve the readability of the presentation and interpretation of the data in our manuscript. Accordingly, single data points can now be seen in the new version of the manuscript.

Thank you.

3. If in the figure legend, it is written n=6 pooled from 2 independent experiments, does this mean n=6 per group, with 2 biological controls and 3 technical replicates (3 stimulated wells/reaction rooms)?

Response:

N=6 pooled from 2 independent experiments means 6 biological replicates (6 independent human monocyte donors), divided between 2 independent experiments (2 different dates, different reagent preparations). This has been clarified in the figure legends of the new version of the manuscript.

Thank you.

4. What does "# vs. control, #,*" mean? I assume # stands for "statistical significant vs. the first column (control)?"

Response:

The assumption of the reviewer is correct. To increase readability, this has been clarified in the figure legends of the new version of the manuscript.

Thank you.

5. Fig.2A: As there are just three data points each and there are no statistical significances, I have to ask why are there no more data points?

Response:

As suggested by the reviewer, we increased the number of biological replicates (Figure 2A and Figure 3B).

Thank you.

6. Fig.2D: I wonder why the animal experiment was not exploited further. Were there 15 mice per group? Could you add the individual curves at least in the supplementals? Was there any further analysis, for example of LPMC? In line 347, the authors claim "we show that FADS2 inhibition decreased both trained immunity and the severity of colitis in a murine model." Where are the data for trained immunity in this colitis model? Apologies if I missed this in the manuscript.

Response:

As suggested by the reviewer, we have added the individual curves of the 15 mice per group (new Figure S4C). Unfortunately, it was not possible to perform any further analysis outside of documentation of mice weight, which we have also added in supplementation (new Figure S4D). In order to decrease the focus on this experiment, Fig 2D in the previous version of the manuscript was moved to Figure S4A, B in the new version of the manuscript.

Thank you. Although it is a shame that the in vivo model has no further data. FACSing monocytes and macrophages from different organs would have been interesting. Or at least put some tissues in formalin to have the opportunity for histopathological analyses at a later time point. Fig 4D has a bug.

We understand how the sentence in line 347 might be misconstrued. We meant that FADS2 inhibition decreases trained immunity in our in vitro model (Figure 2C) and decreases the severity of colitis in the murine model (new Figure S4). This sentence has been restructured in the new version of the manuscript (page 15, line 364/365).

Thank you.

7. Fig. 3A: Isn't Celecoxib a COX2-Inhibitor?

Response:

This mistake has been corrected in the new version of the manuscript.

Thank you.

8. Line 306: "Taking advantage of models," I think there is something missing here.

Response:

This sentence has been corrected in the new version of the manuscript.

Thank you.

9. Fig. 5: figure legends are too short here. I assume for A there were sorted monocytes used in vitro and for B PBMC from the patients. The data for A and B are presented in a very confusing manner. What is the order of the SNPs in the heatmaps? It is very difficult to find or search for the same SNPs over the four heatmaps. Also for the cytokine data, the abbreviations should be

explained (TT, TC, CC, ...).

Response:

Figure 5A pertains to adherent monocytes isolated by density gradient (page 7, line 162), while in Figure 5B indeed PBMCs isolated from BCG vaccinated (prior, 14 days and 90 days after) healthy individuals (page 8, line 167).

Ok.

Regarding the order of the SNPs in the heatmaps, we opted to maintain the order of the genes consistent between heatmaps, as different heatmaps do not necessarily show the same SNPs. The SNPs here presented are the top hits (lowest p-value) in each cytokine training stimulus pair. If there are two SNPs that are in not in linkage disequilibrium (LD), we represented both in the heatmap. It is important to note that the mentioned SNPs are not necessary causal to the phenotype studied, they could be in LD with the causal SNP and still be the most strongly associated (lowest p-value) with the trait (in this case change in cytokine production).

As suggested by the reviewer, to increase readability we have altered the figure and extended the legend to include an explanation of abbreviations.

Ok. Although I still wondering about the meaning of the SNP expression. Do these top hits have known functional relevance?

10. Fig.5A: Do the cytokine data belong to the b-glucan or the BCG samples?

Response:

In Figure 5A, we can observe a heatmap which corresponds to the fold change of TNF and IL-6 upon β -glucan-induced trained immunity (Figure 5A top left panel) and BCG-induced trained immunity (Figure 5A bottom left panel). The boxplots on the right correspond to the SNPs in bold on the heatmaps, which are the one with the lowest pvalues. The boxplot of FADS2 rs9326418 (Figure 5A top right) is in bold in the β -glucan heatmap and the boxplot of LOX12 rs12232535 (Figure 5A bottom right) is in bold in the BCG heatmap. To increase readability, we have modified this figure in the new version of the manuscript.

Thank you.

11. Fig. 5B: Do the cytokine data belong to the Day 14 or day 90 samples? Why was *S. aureus* used? Why working with PBMC instead of myeloid cells (macrophages/monocytes)?

Response:

In Figure 5B we can observe a heatmap which corresponds to the fold change of TNF, IL-6 and IL-1 β between Day 14 and Day 0 (Figure 5B top left panel) and a heatmap that pertains to Day 90 and Day 0 (Figure 5B bottom left panel). The boxplots on the right correspond to the SNPs in bold on the heatmaps, which are the one with the lowest pvalues.

The boxplot of LOX12 rs955461 (Figure 5B top right) is in bold in the Day 14 heatmap and the boxplot of FADS2 rs11600019 (Figure 5B bottom right) is in bold in the Day 90 heatmap. To increase readability, we have modified this figure in the new version of the manuscript.

Thank you.

For the analysis present in Figure 5B, we used a previously established cohort (PMID: 32692728, 36094960). In this cohort *S. aureus* was used as a proxy for heterologous stimulus. The cohort used PBMCs instead of monocytes not only to assess the concentrations of the monocytes derived cytokines, such as IL-6, TNF and IL-1 β , but also to evaluate the potential of the BCG vaccine to enhance T cell-derived cytokines upon a heterologous stimulus. However, in the present study we focused on myeloid derived cytokines.

Ok.

12. Fig 5C: I am not convinced by these data: the placebo always goes up and for 14-HDA it is just because of one "outlier" that this becomes a difference. Can the authors comment on that?

Response:

This is also a fair point raised by the reviewer. The possible seasonal effect is indeed one reason why we included the placebo group in the study. We have adjusted Figure 5C in the new version of the manuscript to show a fold change between V2 and V1 instead of absolute quantification, in order to better illustrate the effect of BCG vaccination. There is no statistical difference between visits in the placebo group. On the other hand, the BCG vaccinated group showed a statistically significant increase of 12-HEPE and 14-HDHA while 12-HETE approached significance.

Thank you for the explanation and the reshaping of the figure. Yet, as the difference should be between Placebo and BCG, I still can't find this convincing. For 14-HDHA I could imagine a tendency, but 12-HETE and 12-HEPE solely depend on one outlier each to show any difference between Placebo and BCG.

13. Line 261: What is the connection between the DSS data and LXRs?

Response:

LXR is a transcription factor that regulates fatty acid homeostasis. Particularly, LXR controls the transcription of FADS2 (PMID: 25838428; 31658997; 28041958). In the present manuscript, we observed that LXR inhibition (as also reported in PMID: 35682840) decreases BCG-enhanced IL-6 and TNF production (new Figure 2D) as does FADS2 inhibition (new Figure 2C). Additionally, we report that FADS2 inhibition in a DSS mouse colitis model decreases disease score (new Figure S4). We have added a sentence to clarify this connection in the new version of the manuscript (Page 11, Line 263-265)

Thank you.

Reviewer #3 (Remarks to the Author):

The authors provide further evidence, which has strengthen the conclusions drawn in the manuscript. I have no further queries.

Response to the Reviewer's Concerns

Reviewer#2

6. Fig.2D: I wonder why the animal experiment was not exploited further. Were there 15 mice per group? Could you add the individual curves at least in the supplementals? Was there any further analysis, for example of LPMC? In line 347, the authors claim "we show that FADS2 inhibition decreased both trained immunity and the severity of colitis in a murine model." Where are the data for trained immunity in this colitis model? Apologies if I missed this in the manuscript.

Response:

As suggested by the reviewer, we have added the individual curves of the 15 mice per group (new Figure S4C). Unfortunately, it was not possible to perform any further analysis outside of documentation of mice weight, which we have also added in supplementation (new Figure S4D). In order to decrease the focus on this experiment, Fig 2D in the previous version of the manuscript was moved to Figure S4A, B in the new version of the manuscript.

Thank you. Although it is a shame that the in vivo model has no further data. FACSing monocytes and macrophages from different organs would have been interesting. Or at least put some tissues in formalin to have the opportunity for histopathological analyses at a later time point.

Fig 4D has a bug.

Response 2:

We thank the reviewer for this comment. However, it is not clear to us what is meant with "bug" in Figure S4D. As far as we can see the weight loss in % as mean \pm SD is depicted as stated in the caption.

9. Fig. 5: figure legends are too short here. I assume for A there were sorted monocytes used in vitro and for B PBMC from the patients. The data for A and B are presented in a very confusing manner. What is the order of the SNPs in the heatmaps? It is very difficult to find or search for the same SNPs over the four heatmaps. Also for the cytokine data, the abbreviations should be explained (TT, TC, CC, ...).

Response:

Figure 5A pertains to adherent monocytes isolated by density gradient (page 7, line 162), while in Figure 5B indeed PBMCs isolated from BCG vaccinated (prior, 14 days and 90 days after) healthy individuals (page 8, line 167).

Ok.

Regarding the order of the SNPs in the heatmaps, we opted to maintain the order of the genes consistent between heatmaps, as different heatmaps do not necessarily show the same SNPs. The SNPs here presented are the top hits (lowest p-value) in each cytokine training stimulus pair. If there are two SNPs that are in not in linkage disequilibrium (LD), we represented both in the heatmap. It is important to note that the mentioned SNPs are not necessary causal to the phenotype studied, they could be in LD with the causal SNP and still be the most strongly associated (lowest p-value) with the trait (in this case change in cytokine production).

As suggested by the reviewer, to increase readability we have altered the figure and extended the legend to include an explanation of abbreviations.

Ok. Although I still wondering about the meaning of the SNP expression. Do these top hits have known functional relevance?

Response 2:

In this study we identified some suggestive quantitative trait loci associated with cytokine production of trained immunity responses. It is important to note that for the cohort studied we did not perform whole-genome sequencing, but instead, the cohort was genotyped. We used a commercially available SNP chip (Infinium Global Screening Array MD v1.0 from Illumina) that allowed us to obtain genotype information on approximately 4 million single-nucleotide polymorphisms by imputation.

The associations found are suggestive of a relationship between variations in the genomic region (the SNP reported is not necessarily the causal variant) and the trait. The assumption that follows is that if a variation in the proximity of a gene is associated with changes of a trait, the gene neighbouring that variant may play a role in the regulation of that trait in the model studied.

Regarding functional roles of the SNPs reported, I could find GWAS studies that reported rs2003019, rs592931 and rs41290534 associated with different phenotypes (Table 1) but could find no other information on the remaining SNPs.

Table 1 Phenotypes from GWAS studies associated with SNPs reported in Fig 5 A,B

SNP	Consequence type	Phenotype, disease and trait	PMID
rs2003019	intron variant	Phosphatidylethanolamine levels	35668104
rs592931	intron variant	Educational attainment	35361970
		Smoking initiation	36477530
rs41290534	regulatory region variant	Nonalcoholic fatty liver disease	35047847

12. Fig 5C: I am not convinced by these data: the placebo always goes up and for 14-HDA it is just because of one "outlier" that this becomes a difference. Can the authors comment on that?

Response:

This is also a fair point raised by the reviewer. The possible seasonal effect is indeed one reason why we included the placebo group in the study. We have adjusted Figure 5C in the new version of the manuscript to show a fold change between V2 and V1 instead of absolute quantification, in order to better illustrate the effect of BCG vaccination. There is no statistical difference between visits in the placebo group. On the other hand, the BCG vaccinated group showed a statistically significant increase of 12-HEPE and 14-HDHA while 12-HETE approached significance.

Thank you for the explanation and the reshaping of the figure. Yet, as the difference should be between Placebo and BCG, I still can't find this convincing. For 14-HDHA I could imagine a tendency, but 12-HETE and 12-HEPE solely depend on one outlier each to show any difference between Placebo and BCG.

Response 2:

We thank the reviewer for this comment and agree. However, we designed this study with the two groups (placebo and BCG) and two time points (V1 and V2), to account for

seasonal/experimental and also interindividual variation. Due to the relatively high interindividual variation, we consider that the best approach would be to probe the differences between visits for either group. Such a paired comparison has more statistical power to detect small differences than the unpaired comparison required to compare Placebo to BCG treated groups. Otherwise we would need to test many more individuals due to statistical power limitations. Unfortunately, it was not possible to include more participants in this analysis.